# Zero-shot protein stability prediction by inverse folding models: a free energy interpretation

**Jes Frellsen**[*][†]
Technical University
of Denmark

**Maher M. Kassem**[*]
Novonesis[‡]

**Tone Bengtsen**
Novonesis[‡]

**Lars Olsen**
Novonesis

**Kresten Lindorff-Larsen**
University of
Copenhagen

**Jesper Ferkinghoff-Borg**
Novo Nordisk

**Wouter Boomsma**[†]
University of
Copenhagen

## Abstract

Inverse folding models have proven to be highly effective zero-shot predictors of protein stability. Despite this success, the link between the amino acid preferences of an inverse folding model and the free-energy considerations underlying thermodynamic stability remains incompletely understood. A better understanding would be of interest not only from a theoretical perspective, but also potentially provide the basis for stronger zero-shot stability prediction. In this paper, we take steps to clarify the free-energy foundations of inverse folding models. Our derivation reveals the standard practice of likelihood ratios as a simplistic approximation and suggests several paths towards better estimates of the relative stability. We empirically assess these approaches and demonstrate that considerable gains in zero-shot performance can be achieved with fairly simple means.

## 1 Introduction

Quantifying how amino acid substitutions affect protein structural stability is of fundamental importance for understanding human genetic diseases (Stein et al., 2019), and for our ability to design and optimise industrial enzymes and protein therapeutics (Listov et al., 2024). In recent years, multiplexed assays of variant effects (MAVE; see, e.g., Starita et al., 2017) experiments have greatly enhanced our ability to characterize variants experimentally, but still only scratch the surface compared to the astronomically large number of possible variants (we use the term variant to refer to a protein that differs from a reference, 'wild-type' protein by one or more amino acid substitutions). *In silico* prediction, therefore, remains an important tool, as a low-cost and fast alternative to experimental characterisation, but also to extrapolate meaningfully from experimental results to uncharacterized variants.

Despite the progress in high-throughput experimental characterisation, variant-effect data still falls in the low-data regime, and supervised learning in this domain has been observed to be prone to overfitting, exemplified by several large-scale studies on human variants (Livesey and Marsh, 2020). As a consequence, unsupervised or weakly supervised methods are often the most attractive approach to the problem. For protein stability prediction, inverse folding models, which provide a probability distribution over amino acid sequences given a fixed 3D structure, have emerged as particularly useful. In particular, variations of log-ratios based on the form

$$-\ln \frac{p(\text{variant sequence} \mid \text{wild-type structure})}{p(\text{wild-type sequence} \mid \text{wild-type structure})} \tag{1}$$

---

[*]Equal contribution. [‡]Work performed while employed at Novonesis.
[†]Corresponding authors: `jefr@dtu.dk` & `wb@di.ku.dk`

39th Conference on Neural Information Processing Systems (NeurIPS 2025).

have shown strong empirical correlation with experimental measurements of stability (Boomsma and Frellsen, 2017; Meier et al., 2021; Hsu et al., 2022; Dutton et al., 2024; Cagiada et al., 2025).

An inverse folding model describes amino acid preferences conditioned on their structural environment. While it is intuitively appealing to assume that likely amino acids correspond to stable protein structures, the formal connection between these probabilities and folding free energies remains incompletely understood. For instance, since thermodynamic stability is an ensemble property (a property of the entire structural distribution), it is not clear why it would be sufficient to condition on a single structure. Likewise, since thermostability is a balance between a folded and an unfolded state, one might reasonably wonder whether one should explicitly model the unfolded-state propensities, in addition to the current practice of generally considering the folded state alone. This raises the question: *Can we interpret these inverse folding models in terms of thermodynamic stability?*

In this paper, we derive a theoretical connection between inverse folding probabilities and thermodynamic stability and use this to elucidate current practices. We also suggest a number of improvements to the current protocols. Our contributions are:

- We derive a formal relationship between changes in thermodynamic stability ($\Delta\Delta G$) and changes in inverse folding probabilities, and describe the approximations necessary to explain the current practice of simple probability ratios, cf. eq. (1).

- We show that the current practice corresponds to single-sample Monte Carlo estimates, and demonstrate empirical performance gains by using multiple samples from molecular dynamics (MD) simulations or the BioEmu generative model (Lewis et al., 2025).

- We show that the unfolded state can be disregarded, cf. eq. (1), but this introduces an additional factor in the probability ratio. We also propose and evaluate several strategies for explicitly modelling the unfolded state and show that these improve empirical performance.

## 2 Background

Assuming isothermal-isobaric conditions (i.e., an NPT ensemble), the stability of a protein is determined as the difference in Gibbs free energy, $\Delta G$, between its folded and unfolded states. These two states are *macro states* in the sense that they each correspond to *a set* of structural conformations. How this quantity changes as a result of changing one or several amino acid residues with another (substitutions) is referred to as a $\Delta\Delta G$. Denoting the original sequence as *wild type* (WT) with amino sequence $\boldsymbol{a}$ and the new sequence as *variant* (MT) with sequence $\boldsymbol{a}'$, we have that

$$\Delta\Delta G_{\boldsymbol{a}\to\boldsymbol{a}'} = \Delta G_{\boldsymbol{a}'}^{\mathrm{U}\to\mathrm{F}} - \Delta G_{\boldsymbol{a}}^{\mathrm{U}\to\mathrm{F}} = (G_{\boldsymbol{a}'}^{\mathrm{F}} - G_{\boldsymbol{a}'}^{\mathrm{U}}) - (G_{\boldsymbol{a}}^{\mathrm{F}} - G_{\boldsymbol{a}}^{\mathrm{U}}). \tag{2}$$

### 2.1 Gibbs free energy

The Gibbs free energy is calculated from the Boltzmann distribution, expressing the probability of a protein microstate with structural degrees of freedom $\boldsymbol{\chi} \in \mathfrak{X}$ and solvent $\boldsymbol{w}$, given by

$$p(\boldsymbol{\chi}, \boldsymbol{w}|\boldsymbol{a}, \beta) = Z_{\boldsymbol{a}}^{-1} e^{-\beta H_{\boldsymbol{a}}(\boldsymbol{\chi}, \boldsymbol{w})} , \; Z_{\boldsymbol{a}} = \int e^{-\beta H_{\boldsymbol{a}}(\boldsymbol{\chi}, \boldsymbol{w})} \, \mathrm{d}\boldsymbol{\chi} \, \mathrm{d}\boldsymbol{w} \tag{3}$$

where $H_{\boldsymbol{a}}(\boldsymbol{\chi}, \boldsymbol{w}) = U_{\boldsymbol{a}}(\boldsymbol{\chi}, \boldsymbol{w}) + PV(\boldsymbol{\chi}, \boldsymbol{w})$ is the enthalpy of the microstate for amino acid sequence $\boldsymbol{a}$, $U_{\boldsymbol{a}}(\boldsymbol{\chi}, \boldsymbol{w})$ is the internal energy (Hamiltonian), $V(\boldsymbol{\chi}, \boldsymbol{w})$ is volume of the microstate, $P$ is the pressure, and $\beta = \frac{1}{k_B T}$ is the inverse of the thermodynamic temperature. The Gibbs free energy associated with the amino acid sequence $\boldsymbol{a}$ is defined as

$$G_{\boldsymbol{a}} = -\beta^{-1} \log Z_{\boldsymbol{a}}. \tag{4}$$

Since our focus is on the degrees of freedom of the protein, we integrate out the solvent degrees of freedom. Furthermore, we split the structure degrees of freedom $\boldsymbol{\chi} = (\boldsymbol{x}, \boldsymbol{s})$ into backbone degrees of freedom $\boldsymbol{x}$ and side-chain degrees of freedom $\boldsymbol{s}$, and integrate out the side chains. The probability of a backbone configuration $\boldsymbol{x}$ for a given amino acid sequence $\boldsymbol{a}$ is then given by

$$p(\boldsymbol{x}|\boldsymbol{a}, \beta) = Z_{\boldsymbol{a}}^{-1} e^{-\beta H_{\boldsymbol{a}}(\boldsymbol{x})} , \; Z_{\boldsymbol{a}} = \int e^{-\beta H_{\boldsymbol{a}}(\boldsymbol{x})} \, \mathrm{d}\boldsymbol{x} \tag{5}$$

where $H_{\boldsymbol{a}}(\boldsymbol{x})$ is the free energy associated with an implicit treatment of the solvent and side-chain freedom defined by $e^{-\beta H_{\boldsymbol{a}}(\boldsymbol{x})} = \int e^{-\beta H_{\boldsymbol{a}}(\boldsymbol{\chi}, \boldsymbol{w})} \, \mathrm{d}\boldsymbol{s} \, \mathrm{d}\boldsymbol{w}$.

## 2.2 Thermodynamic stability

To calculate the stability of a protein, we usually partition[3] the space of structural degrees of freedom $\mathfrak{X}$ into a folded subset $\mathfrak{X}_{\boldsymbol{a}}^{\mathrm{F}}$ and an unfolded subset $\mathfrak{X}_{\boldsymbol{a}}^{\mathrm{U}}$ (Brandts, 1969; Lindorff-Larsen and Teilum, 2021). We will use $S \in \{\mathrm{F, U}\}$ to denote either of the two states. We will assume that the states can be fully characterized by the backbone degrees of freedom, such that the space of backbone degrees of freedom $\mathbb{X}_{\boldsymbol{a}}$ can be partitioned into a folded subset $\mathbb{X}_{\boldsymbol{a}}^{\mathrm{F}}$ and an unfolded subset $\mathbb{X}_{\boldsymbol{a}}^{\mathrm{U}}$. For generality, we use a soft partitioning given by the probability $p(S|\boldsymbol{x}, \boldsymbol{a})$, where the hard partitioning above is a special case specified through the indicator function. We can then write the partition function (normalisation constant) for the state $S$ as

$$Z_{\boldsymbol{a}}^{S} = \int e^{-\beta H_{\boldsymbol{a}}(\boldsymbol{x})} p(S|\boldsymbol{x}, \boldsymbol{a}) \, \mathrm{d}\boldsymbol{x}. \tag{6}$$

Similarly to eq. (4), the free energy of state $S$ is then given by

$$G_{\boldsymbol{a}}^{S} = -\beta^{-1} \log Z_{\boldsymbol{a}}^{S}, \tag{7}$$

and the folding stability of the protein with amino acid sequence $\boldsymbol{a}$ is given by

$$\Delta G_{\boldsymbol{a}}^{\mathrm{U}\to\mathrm{F}} = G_{\boldsymbol{a}}^{\mathrm{F}} - G_{\boldsymbol{a}}^{\mathrm{U}} \quad \text{or equivalently} \quad \beta \Delta G_{\boldsymbol{a}}^{\mathrm{U}\to\mathrm{F}} = \ln \frac{Z_{\boldsymbol{a}}^{\mathrm{U}}}{Z_{\boldsymbol{a}}^{\mathrm{F}}}. \tag{8}$$

If we write the stability in terms of integrals over the Boltzmann distributions, we see that the stability can be expressed as a ratio of probabilities

$$\beta \Delta G_{\boldsymbol{a}}^{\mathrm{U}\to\mathrm{F}} = \ln \frac{\int e^{-\beta H_{\boldsymbol{a}}(\boldsymbol{x})} p(\mathrm{U}|\boldsymbol{x}, \boldsymbol{a}) \mathrm{d}\boldsymbol{x}}{\int e^{-\beta H_{\boldsymbol{a}}(\boldsymbol{x})} p(\mathrm{F}|\boldsymbol{x}, \boldsymbol{a}) \mathrm{d}\boldsymbol{x}} = \ln \frac{\int p(\boldsymbol{x}|\boldsymbol{a}, \beta) p(\mathrm{U}|\boldsymbol{x}, \boldsymbol{a}) \mathrm{d}\boldsymbol{x}}{\int p(\boldsymbol{x}|\boldsymbol{a}, \beta) p(\mathrm{F}|\boldsymbol{x}, \boldsymbol{a}) \mathrm{d}\boldsymbol{x}} = \ln \frac{p(S{=}\mathrm{U}|\boldsymbol{a}, \beta)}{p(S{=}\mathrm{F}|\boldsymbol{a}, \beta)}, \tag{9}$$

where $p(S|\boldsymbol{a}, \beta) = \int p(\boldsymbol{x}|\boldsymbol{a}, \beta) p(S|\boldsymbol{x}, \boldsymbol{a}) \, \mathrm{d}\boldsymbol{x}$ denotes the probability of finding the protein with sequence $\boldsymbol{a}$ in state $S$. Since $p(\mathrm{F}|\boldsymbol{a}, \beta) + p(\mathrm{U}|\boldsymbol{a}, \beta) = 1$, it follows from eq. (9) that

$$\beta \Delta G_{\boldsymbol{a}}^{\mathrm{U}\to\mathrm{F}} = \ln \frac{1 - p(S{=}\mathrm{F}|\boldsymbol{a}, \beta)}{p(S{=}\mathrm{F}|\boldsymbol{a}, \beta)} = \ln \left( \frac{1}{p(S{=}\mathrm{F}|\boldsymbol{a}, \beta)} - 1 \right), \tag{10}$$

which means that knowing $p(S{=}\mathrm{F}|\boldsymbol{a}, \beta)$ is sufficient for calculating the stability $\beta \Delta G_{\boldsymbol{a}}^{\mathrm{U}\to\mathrm{F}}$.

## 2.3 Databases of structure-sequence pairs

Consider a dataset of structure-sequence pairs $D = \{(\boldsymbol{\chi}_i, \boldsymbol{a}_i)\}_i$ that is sampled from the marginal of some data-generating distribution $p_{\mathrm{D}}(\boldsymbol{\chi}, \boldsymbol{a}, \beta)$. The dataset could, for instance, be the Protein Data Bank (PDB; Berman et al., 2000) or some subset of it. We make the fairly strong **assumption** that all structures in the database are sampled approximately from their respective Boltzmann distribution with potentially different and unknown (latent) $\beta$. That is, for any $\boldsymbol{a}$ and $\beta$, we assume

$$p_{\mathrm{D}}(\boldsymbol{\chi}|\boldsymbol{a}, \beta) \approx p(\boldsymbol{\chi}|\boldsymbol{a}, \beta). \tag{11}$$

Note that thermodynamics do not, per se, make any statements about the marginal distribution over the amino sequences. Furthermore, the data marginal over sequences, $p_{\mathrm{D}}(\boldsymbol{a})$, may not reflect the true biological sequence prevalence due to data collection biases, as documented for the PDB (Gerstein, 1998; Orlando et al., 2016).

## 2.4 Inverse folding models

We assume a joint model $p_\theta(\boldsymbol{a}, \boldsymbol{x}, \beta)$ over the sequence $\boldsymbol{a}$, backbone degrees of freedom $\boldsymbol{x}$, and inverse temperature $\beta$. Since available datasets typically do not include the inverse temperature, this limits what can be learned in practice. From a dataset of structure–sequence pairs, we learn the conditional distribution $p_\theta(\boldsymbol{a}|\boldsymbol{x})$, referred to as the *inverse folding model*, as well as the marginal distribution $p_\theta(\boldsymbol{a})$ over sequences. For simplicity, we let $\theta$ denote the combined parameters of all components, although in practice the parameter sets may be disjoint and estimated separately.

Building on the assumption in eq. (11), we further assume that the model provides a good estimate of the conditional distribution over structures given a sequence and inverse temperature, such that

$$p_\theta(\boldsymbol{x}|\boldsymbol{a}, \beta) \approx p_{\mathrm{D}}(\boldsymbol{x}|\boldsymbol{a}, \beta) \approx p(\boldsymbol{x}|\boldsymbol{a}, \beta). \tag{12}$$

This reflects the idea that the learned posterior approximates the Boltzmann distribution well.

---

[3]This means that $\mathfrak{X}_{\boldsymbol{a}} = \mathfrak{X}_{\boldsymbol{a}}^{\mathrm{F}} \cup \mathfrak{X}_{\boldsymbol{a}}^{\mathrm{U}}$ and $\mathfrak{X}_{\boldsymbol{a}}^{\mathrm{F}} \cap \mathfrak{X}_{\boldsymbol{a}}^{\mathrm{U}} = \varnothing$.

# 3 Methods

Consider a wild-type sequence $\boldsymbol{a}$ and a variant sequence $\boldsymbol{a}'$ that differ by one or more amino acid substitutions. The question is now, can we utilise inverse folding models to calculate the change in stability between the two proteins? The definition of the change in stability is given by the difference in folding free energy between the mutant and wild-type. We can reformulate this as

$$\beta\Delta\Delta G_{\boldsymbol{a}\to\boldsymbol{a}'} = \underbrace{\ln\frac{p(S\!=\!\mathrm{U}\,|\boldsymbol{a}',\beta)}{p(S\!=\!\mathrm{F}\,|\boldsymbol{a}',\beta)}}_{\beta\Delta G^{\mathrm{U}\to\mathrm{F}}_{\boldsymbol{a}'}} - \underbrace{\ln\frac{p(S\!=\!\mathrm{U}\,|\boldsymbol{a},\beta)}{p(S\!=\!\mathrm{F}\,|\boldsymbol{a},\beta)}}_{\beta\Delta G^{\mathrm{U}\to\mathrm{F}}_{\boldsymbol{a}}} = \underbrace{\ln\frac{p(S\!=\!\mathrm{U}\,|\boldsymbol{a}',\beta)}{p(S\!=\!\mathrm{U}\,|\boldsymbol{a},\beta)}}_{\beta\Delta\tilde{G}^{\mathrm{U}}_{\boldsymbol{a}'\to\boldsymbol{a}}} - \underbrace{\ln\frac{p(S\!=\!\mathrm{F}\,|\boldsymbol{a}',\beta)}{p(S\!=\!\mathrm{F}\,|\boldsymbol{a},\beta)}}_{\beta\Delta\tilde{G}^{\mathrm{F}}_{\boldsymbol{a}'\to\boldsymbol{a}}}, \tag{13}$$

where the second form expresses the mutation effects on the folded and unfolded states in terms of the two *pseudo* change in free energy terms $\beta\Delta\tilde{G}^{S}_{\boldsymbol{a}'\to\boldsymbol{a}}$, which we define for analysis but are not physical quantities. In the following, we show how these terms can be estimated using importance sampling.

## 3.1 Change in stability using inverse folding models

To calculate the pseudo change in free energy $\Delta\tilde{G}^{S}_{\boldsymbol{a}'\to\boldsymbol{a}}$, we begin by expressing $p(S|\boldsymbol{a}',\beta)$ using importance sampling, similarly to free energy perturbation (Zwanzig, 1954). That is,

$$p(S|\boldsymbol{a}',\beta) = \int p(\boldsymbol{x}|\boldsymbol{a}',\beta)p(S|\boldsymbol{x},\boldsymbol{a}')\,\mathrm{d}\boldsymbol{x} = \mathbb{E}_{\boldsymbol{x}\sim p(\boldsymbol{x}|S,\boldsymbol{a},\beta)}\left[\frac{p(\boldsymbol{x}|\boldsymbol{a}',\beta)p(S|\boldsymbol{x},\boldsymbol{a}')}{p(\boldsymbol{x}|S,\boldsymbol{a},\beta)}\right] \tag{14}$$

Using Bayes' theorem, we can express the structure posterior as $p(\boldsymbol{x}|S,\boldsymbol{a},\beta) = p(\boldsymbol{x}|\boldsymbol{a},\beta)p(S|\boldsymbol{x},\boldsymbol{a})/p(S|\boldsymbol{a},\beta)$, which allows us to rewrite the importance sampling expression from eq. (14) as

$$\frac{p(S|\boldsymbol{a}',\beta)}{p(S|\boldsymbol{a},\beta)} = \mathbb{E}_{\boldsymbol{x}\sim p(\boldsymbol{x}|S,\boldsymbol{a},\beta)}\left[\frac{p(\boldsymbol{x}|\boldsymbol{a}',\beta)p(S|\boldsymbol{x},\boldsymbol{a}')}{p(\boldsymbol{x}|\boldsymbol{a},\beta)p(S|\boldsymbol{x},\boldsymbol{a})}\right]. \tag{15}$$

To evaluate the ratio $\frac{p(\boldsymbol{x}|\boldsymbol{a}',\beta)}{p(\boldsymbol{x}|\boldsymbol{a},\beta)}$, we use the approximation assumption from eq. (12) and apply Bayes' theorem in both numerator and denominator, i.e.,

$$\frac{p(\boldsymbol{x}|\boldsymbol{a}',\beta)}{p(\boldsymbol{x}|\boldsymbol{a},\beta)} \approx \frac{p_\theta(\boldsymbol{x}|\boldsymbol{a}',\beta)}{p_\theta(\boldsymbol{x}|\boldsymbol{a},\beta)} = \frac{p_\theta(\boldsymbol{a}'|\boldsymbol{x},\beta)p_\theta(\boldsymbol{x}|\beta)\,/\,p_\theta(\boldsymbol{a}'|\beta)}{p_\theta(\boldsymbol{a}|\boldsymbol{x},\beta)p_\theta(\boldsymbol{x}|\beta)\,/\,p_\theta(\boldsymbol{a}|\beta)} = \frac{p_\theta(\boldsymbol{a}'|\boldsymbol{x},\beta)}{p_\theta(\boldsymbol{a}|\boldsymbol{x},\beta)}\frac{p_\theta(\boldsymbol{a}|\beta)}{p_\theta(\boldsymbol{a}'|\beta)}. \tag{16}$$

Substituting this into eq. (15), we arrive at

$$\frac{p(S|\boldsymbol{a}',\beta)}{p(S|\boldsymbol{a},\beta)} \approx \mathbb{E}_{\boldsymbol{x}\sim p_\theta(\boldsymbol{x}|S,\boldsymbol{a},\beta)}\left[\frac{p_\theta(\boldsymbol{a}'|\boldsymbol{x},\beta)}{p_\theta(\boldsymbol{a}|\boldsymbol{x},\beta)}\frac{p(S|\boldsymbol{x},\boldsymbol{a}')}{p(S|\boldsymbol{x},\boldsymbol{a})}\right]\frac{p_\theta(\boldsymbol{a}|\beta)}{p_\theta(\boldsymbol{a}'|\beta)}. \tag{17}$$

**Assumptions** Recall that the inverse folding model introduced in section 2.4 does not depend explicitly on $\beta$. To evaluate eq. (17), we assume that the considered $\beta$ is equally representative of both sequences, $\boldsymbol{a}$ and $\boldsymbol{a}'$, and of the structure under our model, i.e., $p_\theta(\beta\mid\boldsymbol{a})\approx p_\theta(\beta\mid\boldsymbol{a}')$ and $p_\theta(\beta\mid\boldsymbol{x},\boldsymbol{a})\approx p_\theta(\beta\mid\boldsymbol{x},\boldsymbol{a}')$. This assumption may be reasonable when the two sequences differ by only a few substitutions. It then follows that

$$\frac{p_\theta(\boldsymbol{a}'|\boldsymbol{x},\beta)}{p_\theta(\boldsymbol{a}|\boldsymbol{x},\beta)} \approx \frac{p_\theta(\boldsymbol{a}'|\boldsymbol{x})}{p_\theta(\boldsymbol{a}|\boldsymbol{x})} \qquad\text{and}\qquad \frac{p_\theta(\boldsymbol{a}'|\beta)}{p_\theta(\boldsymbol{a}|\beta)} \approx \frac{p_\theta(\boldsymbol{a}')}{p_\theta(\boldsymbol{a})}. \tag{18}$$

Furthermore, we assume that a small number of sequence substitutions does not alter the backbone–to–state mapping, that is, $p(S|\boldsymbol{x},\boldsymbol{a})\approx p(S|\boldsymbol{x},\boldsymbol{a}')$, and we can thus assume that the ratio of these terms is $1$ in eq. (17). Together, these assumptions yield

$$\frac{p(S|\boldsymbol{a}',\beta)}{p(S|\boldsymbol{a},\beta)} \approx \mathbb{E}_{\boldsymbol{x}\sim p_\theta(\boldsymbol{x}|S,\boldsymbol{a},\beta)}\left[\frac{p_\theta(\boldsymbol{a}'|\boldsymbol{x})}{p_\theta(\boldsymbol{a}|\boldsymbol{x})}\right]\frac{p_\theta(\boldsymbol{a})}{p_\theta(\boldsymbol{a}')}, \tag{19}$$

providing a means to approximate $\beta\Delta\tilde{G}^{S}_{\boldsymbol{a}'\to\boldsymbol{a}}$ using an inverse folding model.

Note that it would be mathematically tempting to write the importance sampler in eqs. (14) and (15) using the unconditional Boltzmann distribution $p(\boldsymbol{x}|\boldsymbol{a},\beta)$ as the proposal distribution. However, we note that it is much more difficult to sample from $p(\boldsymbol{x}|\boldsymbol{a},\beta)$ than from the conditional $p(\boldsymbol{x}|S,\boldsymbol{a},\beta)$ as it requires sampling both the folded and unfolded states. Unfolding events are typically rare and difficult to sample in, e.g., MD simulations, and we would need to sample multiple folding–unfolding events to obtain a low-variance estimate. If no unfolded structures are sampled, it would (erroneously) imply that $p(\mathrm{F}\mid\boldsymbol{a},\beta)=1$, see appendix A.1 for further discussion.

## 3.2 Change in stability from folded and unfolded ensembles

By combining eqs. (13) and (19), we can express the change in stability as

$$\beta \Delta \Delta G_{\boldsymbol{a} \to \boldsymbol{a}'} \approx \ln \mathbb{E}_{\boldsymbol{x} \sim p_\theta(\boldsymbol{x} \mid \mathrm{U}, \boldsymbol{a}, \beta)} \left[ \frac{p_\theta(\boldsymbol{a}' \mid \boldsymbol{x})}{p_\theta(\boldsymbol{a} \mid \boldsymbol{x})} \right] - \ln \mathbb{E}_{\boldsymbol{x} \sim p_\theta(\boldsymbol{x} \mid \mathrm{F}, \boldsymbol{a}, \beta)} \left[ \frac{p_\theta(\boldsymbol{a}' \mid \boldsymbol{x})}{p_\theta(\boldsymbol{a} \mid \boldsymbol{x})} \right], \qquad (20)$$

which follows from the important observation that the marginal sequence probabilities $p_\theta(\boldsymbol{a})$ and $p_\theta(\boldsymbol{a}')$ cancel between the folded and unfolded terms.

Equation (20) represents a key result: it shows that the change in thermodynamic stability can be estimated using an inverse folding model. The terms inside the expectations can be computed using an inverse folding model, and the full expression becomes tractable through Monte Carlo estimation, provided we can sample structures from the conditional structure distributions $p_\theta(\boldsymbol{x} \mid S, \boldsymbol{a}, \beta)$ for both the unfolded and folded ensembles.

For the folded state, $p_\theta(\boldsymbol{x} \mid \mathrm{F}, \boldsymbol{a}, \beta)$, we can approximate this distribution using structural data available in the dataset. Usually, the dataset will only contain very few structures for each sequence, and if only a single structure $\boldsymbol{x_a}$ is available for sequence $\boldsymbol{a}$, a one-sample approximation yields

$$\mathbb{E}_{\boldsymbol{x} \sim p_\theta(\boldsymbol{x} \mid \mathrm{F}, \boldsymbol{a}, \beta)} \left[ \frac{p_\theta(\boldsymbol{a}' \mid \boldsymbol{x})}{p_\theta(\boldsymbol{a} \mid \boldsymbol{x})} \right] \approx \frac{p_\theta(\boldsymbol{a}' \mid \boldsymbol{x_a})}{p_\theta(\boldsymbol{a} \mid \boldsymbol{x_a})}. \qquad (21)$$

Presumably, a more accurate estimate could be obtained by sampling local structural variations around $\boldsymbol{x_a}$ through molecular simulation. Similarly, simulations could be used to approximate the expectation for the unfolded state. We explore these strategies empirically in section 4.

## 3.3 Change of stability from a folded ensemble or structure

Recall from eq. (10) that knowing $p(\mathrm{F} \mid \boldsymbol{a}, \beta)$ is sufficient for determining the stability of a protein. In this section, we investigate the extent to which we can estimate the change in stability from a folded ensemble alone. We consider two approaches leading to the same expression: in the first case, we assume that $\beta \Delta \tilde{G}^{\mathrm{U}}_{\boldsymbol{a}' \to \boldsymbol{a}} \approx 0$, and in the second case we only consider *ranking* mutations.

### 3.3.1 Simplified change of stability estimation via unfolded-state invariance

If we consider a stable wild-type sequence, i.e., $p(\mathrm{F} \mid \boldsymbol{a}, \beta)$ is close to 1, and a variant with a similar or higher folding probability $p(\mathrm{F} \mid \boldsymbol{a}', \beta)$, then the folded term $\beta \Delta \tilde{G}^{\mathrm{F}}_{\boldsymbol{a}' \to \boldsymbol{a}}$ is close to zero. In this case, the dominant contribution to the stability change $\beta \Delta \Delta G_{\boldsymbol{a} \to \boldsymbol{a}'}$ arises from the unfolded term $\beta \Delta \tilde{G}^{\mathrm{U}}_{\boldsymbol{a}' \to \boldsymbol{a}}$, cf. eq. (20). Thus, for relatively stable variants, the unfolded pseudo-free-energy change, $\beta \Delta \tilde{G}^{\mathrm{U}}_{\boldsymbol{a}' \to \boldsymbol{a}}$, is the key term to evaluate. Conversely, for strongly destabilising mutations, where $p(\mathrm{F} \mid \boldsymbol{a}', \beta) < p(\mathrm{U} \mid \boldsymbol{a}, \beta)$, the dominant term in $\beta \Delta \Delta G_{\boldsymbol{a} \to \boldsymbol{a}'}$ is the folded term $\beta \Delta \tilde{G}^{\mathrm{F}}_{\boldsymbol{a}' \to \boldsymbol{a}}$. These two cases are illustrated in fig. 3. Consequently, if our main interest lies in quantifying strongly destabilising mutations, we may neglect the unfolded state and approximate the stability change as

$$\beta \Delta \Delta G_{\boldsymbol{a} \to \boldsymbol{a}'} \approx -\beta \Delta \tilde{G}^{\mathrm{F}}_{\boldsymbol{a}' \to \boldsymbol{a}} \approx -\ln \mathbb{E}_{\boldsymbol{x} \sim p_\theta(\boldsymbol{x} \mid \mathrm{F}, \boldsymbol{a}, \beta)} \left[ \frac{p_\theta(\boldsymbol{a}' \mid \boldsymbol{x})}{p_\theta(\boldsymbol{a} \mid \boldsymbol{x})} \right] - \ln \frac{p_\theta(\boldsymbol{a})}{p_\theta(\boldsymbol{a}')}, \qquad (22)$$

where the second term accounts for the conditional sequence probabilities under the model. If only a single structure $\boldsymbol{x_a} \sim p_\theta(\boldsymbol{x} \mid \mathrm{F}, \boldsymbol{a})$ is available, we can approximate the expectation with a one-sample estimator

$$\beta \Delta \Delta G_{\boldsymbol{a} \to \boldsymbol{a}'} \approx -\beta \Delta \tilde{G}^{\mathrm{F}}_{\boldsymbol{a}' \to \boldsymbol{a}} \approx -\ln \frac{p_\theta(\boldsymbol{a}' \mid \boldsymbol{x_a})}{p_\theta(\boldsymbol{a} \mid \boldsymbol{x_a})} - \ln \frac{p_\theta(\boldsymbol{a})}{p_\theta(\boldsymbol{a}')}. \qquad (23)$$

The expression in eq. (23) closely resembles standard practice in the field, cf. eq. (1), and thus provides an explanation for zero-shot prediction of inverse-folding models. However, we note that the expression includes an additional correction term that accounts for the frequency of the substituted amino acid under the model (or in the underlying dataset). The fact that the raw log-odds scores work well in practice suggests that this is not a dominating term, but we would expect performance to improve when including it. We investigate this empirically in section 4.

### 3.3.2 Ranking changes of stability

When comparing the stability change of multiple variants $\boldsymbol{a}'^{(1)}$ to $\boldsymbol{a}'^{(n)}$, one would ideally compute and compare their respective values $\beta\Delta\Delta G_{\boldsymbol{a}\to\boldsymbol{a}'^{(i)}}$. However, note that the term $\beta\Delta G_{\boldsymbol{a}}^{\mathrm{U}\to\mathrm{F}}$ is constant across all variants and thus cancels out when comparing values. As a result, ranking variants by their $\beta\Delta G_{\boldsymbol{a}'^{(i)}}^{\mathrm{U}\to\mathrm{F}}$ values preserves the same ordering. Moreover, since $p(\mathrm{F}\,|\,\boldsymbol{a},\beta)$ is constant, $\Delta G_{\boldsymbol{a}'}^{\mathrm{U}\to\mathrm{F}}$ is a monotonic function of $p(\mathrm{F}\,|\,\boldsymbol{a}',\beta)$ and thus also a monotonic function of $-\beta\Delta\tilde{G}_{\boldsymbol{a}'\to\boldsymbol{a}}^{\mathrm{F}}$. See fig. 3 for an illustration and appendix A.2 for a detailed derivation. Therefore, ranking a set of variants $\boldsymbol{a}'^{(1)},\ldots,\boldsymbol{a}'^{(n)}$ by $-\beta\Delta\tilde{G}_{\boldsymbol{a}'^{(i)}\to\boldsymbol{a}}^{\mathrm{F}}$ yields the same ordering as ranking them by their full stability changes $\beta\Delta\Delta G_{\boldsymbol{a}\to\boldsymbol{a}'^{(i)}}$. This implies that if we are only interested in ranking variants, rather than computing exact stability changes, we can ignore the unfolded ensemble and instead use $-\beta\Delta\tilde{G}_{\boldsymbol{a}'^{(i)}\to\boldsymbol{a}}^{\mathrm{F}}$, as given by eqs. (22) and (23).

Importantly, this ranking argument does not rely on the approximation $\beta\Delta\tilde{G}_{\boldsymbol{a}'\to\boldsymbol{a}}^{\mathrm{U}}\approx 0$, but still leads to the same practical expression. This helps explain why strong Spearman correlations have been observed between the simple log-ratio expression $-\ln(p_\theta(\boldsymbol{a}'|\boldsymbol{x}_{\boldsymbol{a}})/p_\theta(\boldsymbol{a}|\boldsymbol{x}_{\boldsymbol{a}}))$ and experimentally measured values of stability changes (Meier et al., 2021), as the Spearman coefficient is a purely rank-based metric.

### 3.4 Change in stability with sequence models

In the previous sections, we derived expressions for estimating the change in thermodynamic stability using inverse folding models. These derivations relied on the ability to sample structural ensembles for both folded and unfolded states. However, in a practical setting, it may be preferable or more convenient to estimate free energy changes using only sequence-based models, without requiring structure-conditioned models.

We assume a joint probabilistic model over amino acid sequences and structural states of the form

$$p_\gamma(\boldsymbol{a},S) = p_\gamma(\boldsymbol{a}\mid S)p_\gamma(S), \tag{24}$$

where $\gamma$ denotes the model parameters. Using Bayes' theorem, the pseudo free energy change for a given structural state $S$ can be expressed as

$$\beta\Delta\tilde{G}_{\boldsymbol{a}'\to\boldsymbol{a}}^{S} = \ln\frac{p(S\mid\boldsymbol{a}',\beta)}{p(S\mid\boldsymbol{a},\beta)} \approx \ln\frac{p_\gamma(S\mid\boldsymbol{a}')}{p_\gamma(S\mid\boldsymbol{a})} = \ln\frac{p_\gamma(\boldsymbol{a}'\mid S)}{p_\gamma(\boldsymbol{a}\mid S)} + \ln\frac{p_\gamma(\boldsymbol{a})}{p_\gamma(\boldsymbol{a}')}, \tag{25}$$

where we assume access to both the marginal and conditional probabilities under the model, and that $p(S\mid\boldsymbol{a},\beta)\approx p_\gamma(S\mid\boldsymbol{a})$ for all $\boldsymbol{a}$. This means that the change in thermodynamic stability can be estimated using only a state-conditional sequence model; see appendix A.3 for further details and section 4 and table 2 for empirical results.

This framework also enables us to combine estimates from different sources by using a sequence-based model for one state and a structure-based model for the other. A practical special case arises when we have a good characterisation of $p_\gamma(\boldsymbol{a}\mid\mathrm{U})$ from data on, e.g., intrinsically disordered proteins or regions, which predominantly represent the unfolded ensemble. By combining this with an inverse folding model for the folded state, and assuming that the marginal sequence probabilities agree across models, i.e., $p_\gamma(\boldsymbol{a})=p_\theta(\boldsymbol{a})$, the change in stability can be expressed as

$$\beta\Delta\Delta G_{\boldsymbol{a}\to\boldsymbol{a}'} \approx \ln\frac{p_\gamma(\boldsymbol{a}'\mid\mathrm{U})}{p_\gamma(\boldsymbol{a}\mid\mathrm{U})} - \ln\mathbb{E}_{\boldsymbol{x}\sim p_\theta(\boldsymbol{x}\mid\mathrm{F},\boldsymbol{a},\beta)}\left[\frac{p_\theta(\boldsymbol{a}'|\boldsymbol{x})}{p_\theta(\boldsymbol{a}|\boldsymbol{x})}\right]. \tag{26}$$

This hybrid approach is particularly useful when only folded structures are available, as a state-conditional sequence model can provide an estimate of the unfolded pseudo–change in stability. We evaluate this hybrid approach empirically in section 4 and include ablations across different combinations of sequence- and structure-based models in table 2.

## 4 Experiments

To evaluate the effects of the different assumptions and approximations discussed in the previous section, we conclude the paper with a series of experiments in which the individual terms are estimated

| | Single sample for folded state | Multiple samples (MD) for folded state |
|---|---|---|
| **Ignoring unfolded state** | $-\ln \dfrac{p_\theta(\mathbf{a}'\vert\mathbf{x_a})}{p_\theta(\mathbf{a}\,\vert\mathbf{x_a})}$ | $-\ln \left\langle \dfrac{p_\theta(\mathbf{a}'\vert\mathbf{x_a})}{p_\theta(\mathbf{a}\,\vert\mathbf{x_a})} \right\rangle_F$ |
| **Ignoring unfolded state. p(a) correction** | $-\ln \dfrac{p_\theta(\mathbf{a}'\vert\mathbf{x_a})}{p_\theta(\mathbf{a}\,\vert\mathbf{x_a})} - \ln \dfrac{p_\theta(\mathbf{a}\,)}{p_\theta(\mathbf{a}')}$ | $-\ln \left\langle \dfrac{p_\theta(\mathbf{a}'\vert\mathbf{x_a})}{p_\theta(\mathbf{a}\,\vert\mathbf{x_a})} \right\rangle_F - \ln \dfrac{p_\theta(\mathbf{a}\,)}{p_\theta(\mathbf{a}')}$ |
| **Unfolded state using Monte Carlo** | $-\ln \dfrac{p_\theta(\mathbf{a}'\vert\mathbf{x_a})}{p_\theta(\mathbf{a}\,\vert\mathbf{x_a})} - \ln \left\langle \dfrac{p_\theta(\mathbf{a}\,\vert\mathbf{x_a})}{p_\theta(\mathbf{a}'\vert\mathbf{x_a})} \right\rangle_U$ | $-\ln \left\langle \dfrac{p_\theta(\mathbf{a}'\vert\mathbf{x_a})}{p_\theta(\mathbf{a}\,\vert\mathbf{x_a})} \right\rangle_F - \ln \left\langle \dfrac{p_\theta(\mathbf{a}\,\vert\mathbf{x_a})}{p_\theta(\mathbf{a}'\vert\mathbf{x_a})} \right\rangle_U$ |
| **Unfolded state using minimal-context inverse-folding** | $-\ln \dfrac{p_\theta(\mathbf{a}'\vert\mathbf{x_a})}{p_\theta(\mathbf{a}\,\vert\mathbf{x_a})} - \ln \dfrac{p_\theta(\mathbf{a_i}\vert\mathbf{x_{a_{i-1:i+1}}})}{p_\theta(\mathbf{a_i'}\vert\mathbf{x_{a_{i-1:i+1}}})}$ | $-\ln \left\langle \dfrac{p_\theta(\mathbf{a}'\vert\mathbf{x_a})}{p_\theta(\mathbf{a}\,\vert\mathbf{x_a})} \right\rangle_F - \ln \dfrac{p_\theta(\mathbf{a_i}\vert\mathbf{x_{a_{i-1:i+1}}})}{p_\theta(\mathbf{a_i'}\vert\mathbf{x_{a_{i-1:i+1}}})}$ |
| **Unfolded state using IDP statistics** | $-\ln \dfrac{p_\theta(\mathbf{a}'\vert\mathbf{x_a})}{p_\theta(\mathbf{a}\,\vert\mathbf{x_a})} - \ln \dfrac{p_\theta(\mathbf{a}\,\vert\mathbf{U_{IDP}})}{p_\theta(\mathbf{a}'\vert\mathbf{U_{IDP}})}$ | $-\ln \left\langle \dfrac{p_\theta(\mathbf{a}'\vert\mathbf{x_a})}{p_\theta(\mathbf{a}\,\vert\mathbf{x_a})} \right\rangle_F - \ln \dfrac{p_\theta(\mathbf{a}\,\vert\mathbf{U_{IDP}})}{p_\theta(\mathbf{a}'\vert\mathbf{U_{IDP}})}$ |

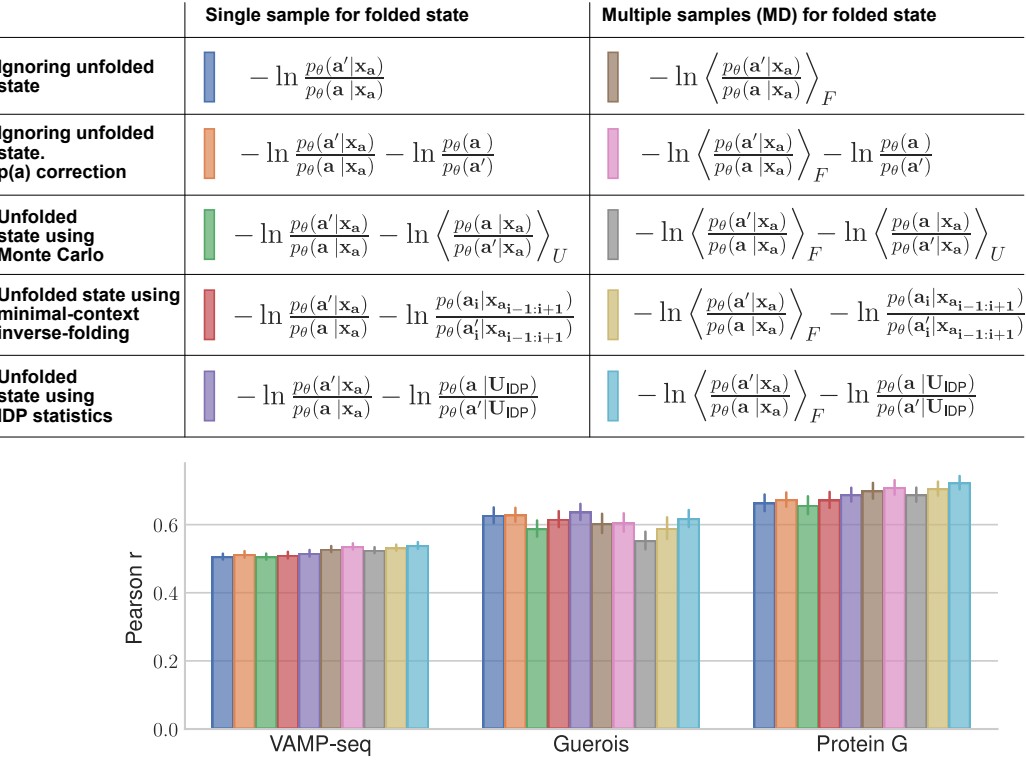

Figure 1: Correlation coefficients obtained using the different expressions discussed in the paper, all involving the inverse-folding model ESM-IF. The top-left expression is the approach typically employed as a zero-shot predictor of protein stability. The left column contains methods that consider only a single folded structure, while the right column considers a structural ensemble from an MD simulation. The different rows represent increasingly accurate approximations to the $\beta\Delta\Delta G$ (see text for details). The error bars represent the standard error of the mean calculated using 100 bootstrap samples. For simplicity, we used the bracket notation $\langle\cdot\rangle_S$ to denote the expectation $\mathbb{E}_{\boldsymbol{x}\sim p_\theta(\boldsymbol{x}|S,\boldsymbol{a},\beta)}[\cdot]$.

using available computational methods on a representative selection of protein datasets. In the following sections, we discuss these choices in turn. We will initially conduct our experiments using the pre-trained ESM inverse folding (ESM-IF) model (Hsu et al., 2022), as it has been shown to perform well in a zero-shot setting (Notin et al., 2023). An ablation with ProteinMPNN is discussed subsequently.

## 4.1 Data

Our primary analysis considers three different data sets. The first is a high-quality data set measuring the thermodynamic stability of nearly all variants of a single 56-residue protein, the B1 domain of Protein G (hereafter called Protein G; Nisthal et al., 2019). The second is an older benchmark set, compiled for the FoldX prediction method by Guerois et al. (2002), mostly consisting of entries from the ProTherm database (Gromiha et al., 1999). This set also directly provides experimental changes in thermodynamic stability, but is heterogeneous in terms of experimental conditions and is known to be biased towards substitutions in which a large amino acid residue is replaced by a smaller one and, in particular, mutations to Alanine (Stein et al., 2019). Finally, we include data generated using variant abundance by massively parallel sequencing (VAMP-seq) experiments that probe stability only indirectly, by quantifying the variant abundance in cultured cells using a combination of fluorescent tags and sequencing (Matreyek et al., 2018). The data generated by VAMP-seq have previously been shown to correlate with both biophysical measurements (Matreyek et al., 2018) and computational predictions (Cagiada et al., 2021) for protein stability, despite the assay reflecting cellular factors beyond thermodynamic stability. We will refer to these sets as Protein G, Guerois, and VAMP-seq, respectively.

We will conclude the experiments section with a brief discussion of scaling. For this purpose, we employ a subset of the mega-scale stability dataset (Tsuboyama et al., 2023) as included in Protein-inGym (Notin et al., 2023). Both VAMP-seq and the mega-scale set represent recent developments in high-throughput assays based on deep sequencing, which are becoming increasingly common for variant characterisation. Combined, the chosen data sets are selected to reflect the different quality/noise regimes in experimental stability data (see appendix B.1 for details).

## 4.2   The folded ensemble

We approximate the expectation over $p_\theta(\boldsymbol{x}|\,\mathrm{F}, \boldsymbol{a}, \beta)$ using unbiased molecular dynamics (MD) simulations. Using the OpenMM framework (Eastman et al., 2017), 20 ns simulations were conducted at 300 K using 2 femtosecond time steps with the Langevin integrator, combined with the Amber 14 force field with a TIP3P water model, adding counter ions to assure overall neutrality. See appendix B.2 for details on the choice of simulation ensemble.

When considering only the folded state, a sequence correction factor arises in eq. (23). For simplicity, we will in our experiments use a position independent sequence model estimated from $p_\mathrm{D}$, meaning that for a single mutation amino acid $i$ the factor becomes $\ln(p_\theta(\boldsymbol{a})/p_\theta(\boldsymbol{a}')) = \ln(p_\theta(\boldsymbol{a}_i)/p_\theta(\boldsymbol{a}_i'))$. We also did the analysis using the ESM2 language model (Lin et al., 2022), but found this choice to be detrimental (see section 4.5 and table 2).

## 4.3   The unfolded ensemble

For the unfolded ensemble, it is less clear what the best approach is, and we therefore try different strategies. In the first approach, we conduct Metropolis-Hastings simulations in the Phaistos framework (Boomsma et al., 2013), using the TorusDBN (Boomsma et al., 2008, 2014) and Basilisk (Harder et al., 2010) statistical models to obtain reasonable backbone and side-chain conformations, but otherwise keep the chain in an unfolded state. We simulated segments with five flanking amino acids on each side of the position of interest, running for 10,000 iterations, where each 100th structure was saved. In the second approach, we again evaluate the ESM-IF model on segments, but unlike the first approach, we now extract a single fixed segment from the crystal structure (i.e., the folded state). The fragment length is kept short (1 flanking amino acid to each side) to approximate an unfolded state, and no structural averaging is done. This approach is similar to that introduced by Dutton et al. (2024), but using segments of length 3 instead of 1. The third approach differs from the first two by not considering the unfolded structural ensemble at all, and instead using a sequence model as detailed in eq. (26). Specifically, we consider protein disorder as a proxy for the unfolded state, and estimate $p_\gamma(\boldsymbol{a}|\,\mathrm{U_{IPD}})$ using amino acid frequencies obtained from disordered regions according to the 'curated-disorder-uniprot' annotation in the MobiDB (Piovesan et al., 2021) database (extracted Jan 21, 2021).

## 4.4   Ablations and scaling

Ablations were performed on the Protein G dataset. First, we repeated all analyses using the ProteinMPNN model (Dauparas et al., 2022). Second, we probed whether the expensive MD simulations could be replaced by predicted ensembles using the BioEmu model (Lewis et al., 2025). After observing promising results, the BioEmu approach was repeated on the larger mega-scale stability dataset (Tsuboyama et al., 2023).

## 4.5   Results

Results for folded/unfolded strategies are reported in fig. 1 and appendix B.1 with additional results on sequence models in table 2. The ProteinMPNN and BioEmu results are found in table 3, while the mega-scale results are displayed in fig. 2.

**Folded state: single-sample vs multi-sample approximation**   The left column in the legend in fig. 1 corresponds to methods that use only a single native structure to approximate the expectation, while the right column approximates the ensemble average using multiple sampled structures. For the VAMP-seq and Protein-G dataset, this choice consistently improved performance. On the Guerois dataset, no general trends can be observed, and we observe extensive fluctuations among the 40 different structures in the dataset, reflecting the heterogeneous nature of this older dataset, and perhaps indicating issues with our MD simulations for some of these systems (see also fig. 4).

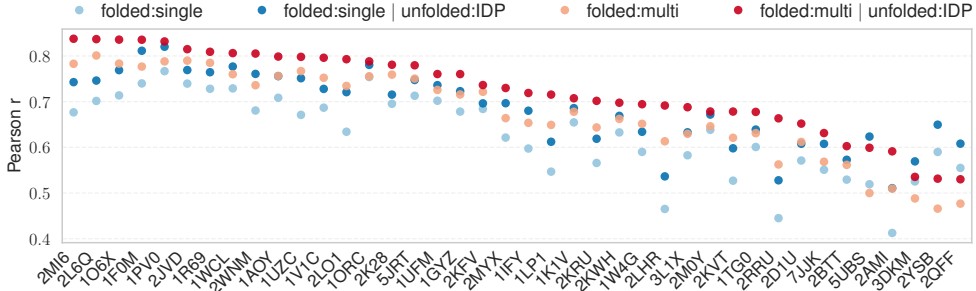

Figure 2: Scaling using BioEmu (Lewis et al., 2025). When replacing MD simulations of the folded state with structural ensembles generated using BioEmu (20 samples; filtered to include only folded states), we still observe consistent improvements over the single-structure performance on this subset of the mega-scale data set (Tsuboyama et al., 2023). This suggests that learned generators of molecular ensembles offer a promising route to scaling our ensemble-based approach in practice.

**Considering only the folded state: the $p_\theta(\mathbf{a})$ correction** For the folded-only case, the experiments generally show an improvement of including the $\ln(p_\theta(\boldsymbol{a})/p_\theta(\boldsymbol{a}'))$ correction term, but the effect is fairly minor as anticipated. We note that in light of eq. (26), the correction term in eq. (23) can be interpreted as a specific choice of model of the unfolded state, namely the one where $p_\gamma(\boldsymbol{a}|\,\mathrm{U})$ is assumed to factorize over positions and follow the general amino acid propensities. This perspective provides another reason why this simplistic model might not provide very accurate results.

**Three models for the unfolded state** Estimating the contribution from the unfolded state using a Monte Carlo simulation worked less well than expected, generally performing worse than the simple log-odds baseline. One explanation could be that ESM-IF has been trained on structures generated by AlphaFold, and thus has learnt specific geometric features that may not be present in the structures generated by our Monte Carlo simulations. Another explanation could be that the sequence- and local structure signal in ESM-IF dominates when no structural environment is present. Since our Monte Carlo sampler uses a proposal distribution that guarantees native-like local structure, ESM-IF apparently displays folded-like preferences when evaluated on unfolded fragments with native local structure. Replacing the Monte Carlo simulation with a single short structural fragment extracted from the folded state yields some improvements, in line with previous reporting (Dutton et al., 2024). Remarkably, the best approach was to approximate the unfolded state using amino acid frequency statistics from disordered regions. Despite its simplicity and ease of implementation, it generally outperforms the other models of the unfolded state on the considered datasets.

**Ablation and scaling** The ablation results in table 3 confirm earlier reports that the likelihoods from ProteinMPNN are less informative for zero-shot stability prediction than those from ESM-IF (Notin et al., 2023). It has been shown that the autoregressive order of evaluation can improve likelihoods Dutton et al. (2024), but this was not implemented here. Despite the different baseline performance levels, both models improve similarly across the different strategies. The BioEmu ablation shows only a very minor drop in performance relative to the MD ensembles, suggesting that BioEmu can serve as an efficient proxy when averaging structural ensembles. When applied to a range of proteins from the mega-scale dataset, the BioEmu protocol indeed yields consistent improvements over the baseline at a fraction of the computational cost of a full MD analysis (fig. 2).

## 5 Related work

The idea of connecting free energies to the statistical properties of experimentally determined protein structures has a long history (Tanaka and Scheraga, 1976; Lifson and Levitt, 1979; Sippl, 1990; Miyazawa and Jernigan, 1996; Simons et al., 1999). Before the advent of deep models, these approaches have essentially been based on deriving approximate free-energy stability energies from single and pairwise amino-acid statistics (Borg et al., 2012). Commonly used physico-chemical based modelling tools for calculating $\Delta\Delta G$ include the FoldX (Schymkowitz et al., 2005) and Rosetta

force-fields (Alford et al., 2017). While providing reasonable estimates for conservative mutations, these force-fields are known to be sensitive to the specific choice of the backbone template in a given application and, as such, do not yield consistent $\Delta\Delta G$ estimates when applied to the full native ensemble of backbone structures. The connection between $\Delta\Delta G$ and inverse-folding likelihoods was initially explored by Boomsma and Frellsen (2017) in the context of a 3D convolutional model. This study introduced a correction term compensating for the base frequencies of amino acids similar to eq. (23), but argued for it in terms of the unfolded state, while we show here that it follows more naturally as a consequence of assuming zero contribution from the unfolded state. More recently, a study demonstrated performance gains by including a correction term by evaluating an inverse-model only on the coordinates of the amino acid in question, motivating it as a representation of the unfolded state (Dutton et al., 2024). Our paper provides the theoretical basis for this argument, and we include a very similar strategy in our experiments (using fragments of length 3). Finally, contemporaneously with our work, a recent study on binding affinity prediction reported substantial performance gains by explicitly incorporating the unfolded state (Jiao et al., 2024), using Bayes' theorem in a similar way as we do in eq. (15), but without considering the full structural ensembles as we do here. Deng et al. (2025) subsequently extended this approach by fine-tuning the inverse folding model on folding stability data.

## 6    Discussion

Log-odds scores from protein inverse-folding models correlate remarkably well with changes in protein stability, but the underlying reasons for this correspondence have remained incompletely understood. In this paper, we take steps to establish a formal connection between the two. We demonstrate that the standard log-odds practice arises as a consequence of a specific set of assumptions, and explore how these assumptions can be relaxed to improve zero-shot prediction further.

Based on our experiments, two choices appear to have the most significant impact over the simple log-odds baseline: 1) including a contribution from the unfolded ensemble, and 2) approximating the structural ensemble of the folded state with more than a single sample. From a practical perspective, both are potentially inconvenient in that they involve molecular simulation. Fortunately, our experiments indicate that the unfolded state can be approximated by a simple static distribution extracted from disordered regions, and we show that computationally-convenient proxies can also be found for the folded ensemble, based on recent generative models of molecular ensembles (Lewis et al., 2025). We therefore anticipate that these improvements can be readily implemented on top of any existing pre-trained free energy model. Finally, we note that while our study has focused on protein stability, it should extend directly to the analysis of binding affinity, generalizing the approach derived by Jiao et al. (2024).

**Limitations**    While our derivations are general, the experiments section necessitates choices regarding practical implementations of the individual terms. We believe we have made reasonable choices, but have not exhaustively explored the space of possible models. We consider our experimental section as a proof-of-concept, exemplifying that a better theoretical treatment *can* lead to gains in performance. The relative size of these performance gains will depend on the protein and the models used to approximate the terms, and cannot be conclusively established from our limited set of experiments. Another outstanding issue is that our analysis does not explain the recent observation that inverse-folding likelihoods also correlate surprisingly well with *absolute* stabilities (Cagiada et al., 2025).

**Broader impact**    As machine learning models play an increasing role in science, it is important to understand how such models work and interact with existing interpretable models. By establishing a link between pre-trained protein models and the free-energy considerations that drive our physical understanding of protein stability, we hope to make these models more broadly applicable to the scientific community. Although we acknowledge that inverse-folding models can be considered dual-use technologies, we believe any such risks are mitigated by the fact that our work focuses on a theoretical understanding of an existing model class, rather than the development of new predictive capabilities.

**Availability**    Code for reproducing the experiments is accessible at: `https://github.com/MachineLearningLifeScience/inverse_folding_free_energies`

## Acknowledgments and Disclosure of Funding

JF and WB are supported by the Center for Basic Machine Learning Research in Life Science (MLLS) through the Novo Nordisk Foundation (NNF20OC0062606), and the Pioneer Centre for AI (DNRF grant P1). KL-L is supported by the PRISM (Protein Interactions and Stability in Medicine and Genomics) centre funded by the Novo Nordisk Foundation (NNF18OC0033950).

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

# A  Details on evaluating the change in stability using inverse folding models

## A.1  Using the unconditional Boltzmann distribution as the proposal

In eq. (15), we use the conditional Boltzmann distribution $p(\boldsymbol{x}|S, \boldsymbol{a}, \beta)$ as the proposal, since it is easier to sample from than $p(\boldsymbol{x}|\boldsymbol{a}, \beta)$. Here, we will investigate the unconditional proposal distribution.

We can write the importance sampler from eq. (14) using the unconditional Boltzmann distribution as

$$p(S|\boldsymbol{a}', \beta) = \int p(\boldsymbol{x}|\boldsymbol{a}')p(S|\boldsymbol{x}, \boldsymbol{a}', \beta)\, \mathrm{d}\boldsymbol{x} = \mathbb{E}_{\boldsymbol{x}\sim p(\boldsymbol{x}|\boldsymbol{a}, \beta)}\left[\frac{p(\boldsymbol{x}|\boldsymbol{a}', \beta)p(S|\boldsymbol{x}, \boldsymbol{a}')}{p(\boldsymbol{x}|\boldsymbol{a}, \beta)}\right]. \tag{27}$$

In the extreme case, when we try to sample from $p(\boldsymbol{x}|\boldsymbol{a}, \beta)$, we efficiently only sample the folded state, i.e., we use $p(\boldsymbol{x}|\mathrm{F}, \boldsymbol{a}, \beta)$ as the proposal distribution. This corresponds to assuming that $p(S|\boldsymbol{a}', \beta) = \mathbb{E}_{\boldsymbol{x}\sim p(\boldsymbol{x}|\mathrm{F}, \boldsymbol{a}, \beta)}\left[\frac{p(\boldsymbol{x}|\boldsymbol{a}', \beta)p(S|\boldsymbol{x}, \boldsymbol{a}')}{p(\boldsymbol{x}|\boldsymbol{a}, \beta)}\right]$. Using Bayes' rule on this assumption gives us that

$$p(S|\boldsymbol{a}', \beta) = \mathbb{E}_{\boldsymbol{x}\sim p(\boldsymbol{x}|\boldsymbol{a}, \beta)}\left[\frac{p(\boldsymbol{x}|\boldsymbol{a}', \beta)p(S|\boldsymbol{x}, \boldsymbol{a}')}{p(\boldsymbol{x}|\boldsymbol{a}, \beta)}\frac{p(\mathrm{F}|\boldsymbol{x}, \boldsymbol{a})}{p(\mathrm{F}|\boldsymbol{a}, \beta)}\right] \tag{28}$$

$$= \frac{1}{p(\mathrm{F}|\boldsymbol{a}, \beta)}\mathbb{E}_{\boldsymbol{x}\sim p(\boldsymbol{x}|\boldsymbol{a}, \beta)}\left[\frac{p(\boldsymbol{x}|\boldsymbol{a}', \beta)p(S|\boldsymbol{x}, \boldsymbol{a}')}{p(\boldsymbol{x}|\boldsymbol{a}, \beta)}p(\mathrm{F}|\boldsymbol{x}, \boldsymbol{a})\right] \tag{29}$$

$$\leq \frac{1}{p(\mathrm{F}|\boldsymbol{a}, \beta)}\mathbb{E}_{\boldsymbol{x}\sim p(\boldsymbol{x}|\boldsymbol{a}, \beta)}\left[\frac{p(\boldsymbol{x}|\boldsymbol{a}', \beta)p(S|\boldsymbol{x}, \boldsymbol{a}')}{p(\boldsymbol{x}|\boldsymbol{a}, \beta)}\right] = \frac{p(S|\boldsymbol{a}')}{p(\mathrm{F}|\boldsymbol{a})}, \tag{30}$$

which is only true for $p(\mathrm{F}|\boldsymbol{a}, \beta) = 1$. So the assumption implies that $p(\mathrm{F}|\boldsymbol{a}, \beta) = 1$.

## A.2  Monotonicity of stability

We aim to show that $\Delta G_{\boldsymbol{a}'}^{\mathrm{U}\to\mathrm{F}}$ is a monotone increasing function of $-\beta\Delta\tilde{G}_{\boldsymbol{a}'\to\boldsymbol{a}}^{\mathrm{F}}$. In the following, all distributions are conditioned on $\beta$, but we omit it to simplify notation. Starting from eq. (10), we can write

$$\beta\Delta G_{\boldsymbol{a}'}^{\mathrm{U}\to\mathrm{F}} = \ln\left(\frac{1}{p(S{=}\mathrm{F}|\boldsymbol{a}')} - 1\right) \tag{31}$$

$$= \ln\left(\frac{1}{\frac{p(S{=}\mathrm{F}|\boldsymbol{a})}{p(S{=}\mathrm{F}|\boldsymbol{a})}p(S{=}\mathrm{F}|\boldsymbol{a}')} - 1\right) \tag{32}$$

$$= \ln\left(\frac{1}{p(S{=}\mathrm{F}|\boldsymbol{a})\exp\left(\ln\frac{p(S{=}\mathrm{F}|\boldsymbol{a}')}{p(S{=}\mathrm{F}|\boldsymbol{a})}\right)} - 1\right) \tag{33}$$

$$= \ln\left(\frac{\exp\left(-\ln\frac{p(S{=}\mathrm{F}|\boldsymbol{a}')}{p(S{=}\mathrm{F}|\boldsymbol{a})}\right)}{p(S{=}\mathrm{F}|\boldsymbol{a})} - 1\right) \tag{34}$$

$$= \ln\left(\frac{\exp(y)}{p(S{=}\mathrm{F}|\boldsymbol{a})} - 1\right) =: f(y) \tag{35}$$

where $y = -\beta\Delta\tilde{G}_{\boldsymbol{a}'\to\boldsymbol{a}}^{\mathrm{F}} = -\ln\frac{p(S{=}\mathrm{F}|\boldsymbol{a}')}{p(S{=}\mathrm{F}|\boldsymbol{a})}$. We note that $y \mapsto \exp(y)$ is monotonically increasing and $z \mapsto \ln(z/c - 1)$ is monotonically increasing for $c > 0$. Since $p(S{=}\mathrm{F}|\boldsymbol{a})$ is a positive constant and function composition preserves monotonicity, we have that $f(y)$ is a monotonically increasing function of $y$.

## A.3  Details on using sequence models for change in stability

By combining eq. (13) with eq. (25), the change in thermodynamic stability can be approximated as

$$\beta\Delta\Delta G_{\boldsymbol{a}\to\boldsymbol{a}'} \approx \ln\frac{p_\gamma(\boldsymbol{a}' \mid \mathrm{U})}{p_\gamma(\boldsymbol{a} \mid \mathrm{U})} - \ln\frac{p_\gamma(\boldsymbol{a}' \mid \mathrm{F})}{p_\gamma(\boldsymbol{a} \mid \mathrm{F})}, \tag{36}$$

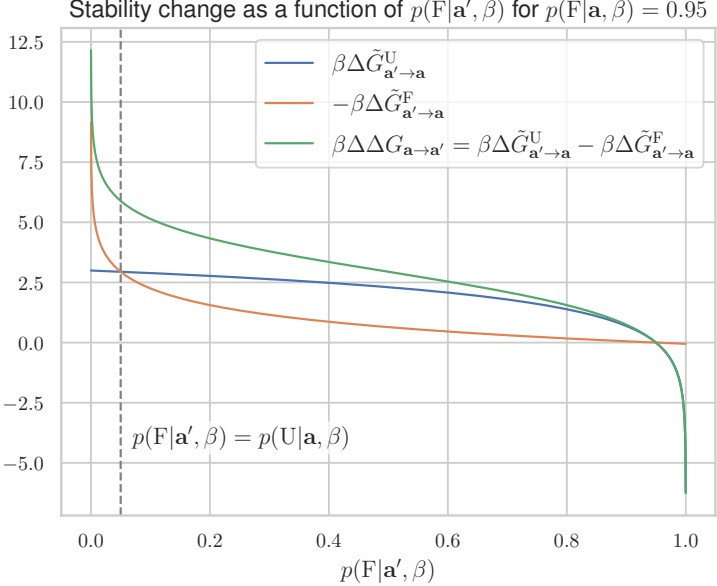

Figure 3: Stability change calculated from the two pseudo–free-energy terms, $\beta\Delta\tilde{G}^{S}_{\boldsymbol{a}'\to\boldsymbol{a}}$, plotted as a function of $p(\mathrm{F}\,|\boldsymbol{a}',\beta)$ for $p(\mathrm{F}\,|\boldsymbol{a},\beta) = 0.95$. As discussed in section 3.3.1, we observe that when $p(\mathrm{F}\,|\boldsymbol{a}',\beta)$ is around 0.95, the unfolded-state term, $\beta\Delta\tilde{G}^{\mathrm{U}}_{\boldsymbol{a}'\to\boldsymbol{a}}$, provides the dominant contribution to the stability change, $\beta\Delta\Delta G_{\boldsymbol{a}\to\boldsymbol{a}'}$, whereas for $p(\mathrm{F}\,|\boldsymbol{a}',\beta) < p(\mathrm{U}\,|\boldsymbol{a},\beta)$, the folded-state term, $\beta\Delta\tilde{G}^{\mathrm{F}}_{\boldsymbol{a}'\to\boldsymbol{a}}$, dominates. We also observe that the stability change, $\beta\Delta\Delta G_{\boldsymbol{a}\to\boldsymbol{a}'}$, is a monotonic function of the variant's folding probability, $p(\mathrm{F}\,|\boldsymbol{a}',\beta)$, as discussed in section 3.3.2 and appendix A.2.

where the marginal sequence probabilities $p_{\gamma}(\boldsymbol{a})$ and $p_{\gamma}(\boldsymbol{a}')$ cancel between the folded and unfolded terms, analogous to the cancellation in eq. (20). This expression shows that we can estimate the change in stability using only a sequence model that provides conditional likelihoods given the structural state, i.e., a model capable of computing $p_{\gamma}(\boldsymbol{a} \mid S)$ for $S \in \{\mathrm{F}, \mathrm{U}\}$.

# B  Experimental details

## B.1  Dataset preprocessing details

The Guerois data set (Guerois et al., 2002) contains 988 entries that we filtered to contain only single amino acid substitutions (i.e. no double and triple substitutions). 911 entries remained after filtering and are associated to 40 PDB structures.

The Protein G data set (Nisthal et al., 2019) contains 907 entries associated with a single PDB (PDBID:1PGA). We used the values labelled as "ddG(mAvg)_mean" which are associated with the lowest median uncertainty reported to be 0.1 kcal/mol. For 107 of the very destabilising entries, only a lower bound of 4.0 kcal/mol were reported. Note that these values are the $\Delta\Delta G$ of unfolding, therefore, we inverted the sign to obtain $\Delta\Delta G$ values for folding.

The VAMP-seq data (Matreyek et al., 2018) contains 8096 entries for two proteins, TPMT and PTEN (associated with two structures with PDBID: 2H11 and 1D5R, respectively). After filtering to include only amino acid residues that are resolved in the protein structures, 6909 entries remain. We use the values labelled as "score" with a negative sign.

For our last experiment, we considered a subset of the mega-scale thermodynamic folding stability dataset (Tsuboyama et al., 2023), with a total of 42253 variants across 40 proteins (PDB-IDs can be found in Figure 2). The experimental values were retrieved through the ProteinGym (Notin et al., 2023) interface without further processing. The structural ensembles for the wild-type variants of these proteins were retrieved from the supporting material of the BioEmu paper (Lewis et al., 2025).

## B.2 Choice of simulation ensemble

For the molecular dynamics simulations used in our study, we initially conducted simulations in an NVT ensemble. Over the course of the study, we refined this protocol and switched to an NPT simulation setup, which we used for the simulations for the Protein G, TPMT and PTEN in the VAMP-seq and Protein G datasets. Since the change in protocol turned out to have very minor effect on the prediction accuracy we decided not to rerun the simulations for the 40 proteins in the Guerois data set.

## B.3 BioEmu samples

The BioEmu model produces both folded and unfolded conformations. Since we only use BioEmu as a proxy for the folded state in our analysis, we filter out conformations with a fraction of native contacts (FNC) below 0.5. Since BioEmu aims to sample states proportionally to their Boltzmann weights, and the considered proteins are stable, less than 1% of samples were discarded by this procedure.

## B.4 Resources

The experiments in this paper comprise running Monte Carlo and molecular dynamics simulations for 40 proteins, in addition to model evaluation of pretrained models on all samples. Since no training was involved, no large scale GPU-resources were necessary for this study.

# C Licenses and references for used assets

Below we list the external software and dataset assets used in this study, along with their licenses and access information:

**ESM-IF** The ESM-IF inverse folding model (Hsu et al., 2022) is released under the MIT license. Source code and pretrained models are available at: https://github.com/facebookresearch/esm

**Phaistos Framework** The Phaistos framework (Boomsma et al., 2013), used for Monte Carlo simulations of unfolded structures, is available under the LGPLv2 or GPLv3 license. The source code is available at: https://sourceforge.net/projects/phaistos

**OpenMM** The OpenMM molecular dynamics engine (Eastman et al., 2017) was used for MD simulations. It is released under a mix of licenses, including MIT, LGPLv3, CC BY 3.0 and several other linces for specific parts (see details at https://github.com/openmm/openmm/blob/master/docs-source/licenses/Licenses.txt) and available at: https://github.com/openmm/openmm

**Protein G Dataset** The thermodynamic stability measurements for the B1 domain of Protein G by Nisthal et al. (2019) is available with no license from: https://www.protabank.org/study_analysis/3xESLyS9/

**Guerois Benchmark Set** This benchmark set of experimental $\Delta\Delta G$ values was compiled by Guerois et al. (2002), with data sourced from the ProTherm database (Gromiha et al., 1999). The dataset is available through the ProTherm database at: https://web.iitm.ac.in/bioinfo2/prothermdb

**VAMP-seq Dataset** Stability-related data for the TPMT and PTEN proteins were taken from Matreyek et al. (2018) and are available for non-profit, non-commercial use at: https://github.com/FowlerLab/VAMPseq

**MobiDB Disorder Statistics** Amino acid frequencies for disordered regions were extracted from the 'curated-disorder-uniprot' entries in the MobiDB database (Piovesan et al., 2021). The database is available under the CC BY 4.0 licence at: https://mobidb.bio.unipd.it

**Mega-scale thermodynamic folding stability dataset** The mega-scale thermodynamic folding stability dataset (Tsuboyama et al., 2023) was accessed through the ProteinGym interface (Notin et al., 2023). It is licenced under the MIT licence.

Table 1: Tabular overview of the Pearson correlation coefficients presented in the fig. 1, where the numbers in parentheses are the standard error of the mean.

| Strategy | | Guerois | Protein G | VAMP-seq |
|---|---|---|---|---|
| *folded-only, single-sample* | $-\ln \frac{p_\theta(\mathbf{a}'\mid\mathbf{x_a})}{p_\theta(\mathbf{a}\mid\mathbf{x_a})}$ | 0.63 (0.02) | 0.66 (0.02) | 0.51 (0.01) |
| *folded-only, $p_\theta(\mathbf{a})$ correction* | $-\ln \frac{p_\theta(\mathbf{a}'\mid\mathbf{x_a})}{p_\theta(\mathbf{a}\mid\mathbf{x_a})} - \ln \frac{p_\theta(\mathbf{a})}{p_\theta(\mathbf{a}')}$ | 0.63 (0.02) | 0.67 (0.02) | 0.51 (0.01) |
| *folded single-sample, unfolded multi-sample (MC)* | $-\ln \frac{p_\theta(\mathbf{a}'\mid\mathbf{x_a})}{p_\theta(\mathbf{a}\mid\mathbf{x_a})} - \ln \left\langle \frac{p_\theta(\mathbf{a}\mid\mathbf{x_a})}{p_\theta(\mathbf{a}'\mid\mathbf{x_a})} \right\rangle_U$ | 0.59 (0.02) | 0.66 (0.02) | 0.51 (0.01) |
| *folded single-sample, inv-fold single-aa as unfolded* | $-\ln \frac{p_\theta(\mathbf{a}'\mid\mathbf{x_a})}{p_\theta(\mathbf{a}\mid\mathbf{x_a})} - \ln \frac{p_\theta(\mathbf{a_i}\mid\mathbf{x_{a_{i-1:i+1}}})}{p_\theta(\mathbf{a_i'}\mid\mathbf{x_{a_{i-1:i+1}}})}$ | 0.62 (0.02) | 0.67 (0.02) | 0.51 (0.01) |
| *folded single-sample, IDP aa-stats as unfolded* | $-\ln \frac{p_\theta(\mathbf{a}'\mid\mathbf{x_a})}{p_\theta(\mathbf{a}\mid\mathbf{x_a})} - \ln \frac{p_\theta(\mathbf{a}\mid\mathbf{U}_{\mathrm{IDP}})}{p_\theta(\mathbf{a}'\mid\mathbf{U}_{\mathrm{IDP}})}$ | 0.64 (0.02) | 0.69 (0.02) | 0.52 (0.01) |
| *folded-only, multi-sample* | $-\ln \left\langle \frac{p_\theta(\mathbf{a}'\mid\mathbf{x_a})}{p_\theta(\mathbf{a}\mid\mathbf{x_a})} \right\rangle_F$ | 0.6 (0.03) | 0.7 (0.03) | 0.53 (0.01) |
| *folded only, multi-sample (MD), $p_\theta(\mathbf{a})$ correction* | $-\ln \left\langle \frac{p_\theta(\mathbf{a}'\mid\mathbf{x_a})}{p_\theta(\mathbf{a}\mid\mathbf{x_a})} \right\rangle_F - \ln \frac{p_\theta(\mathbf{a})}{p_\theta(\mathbf{a}')}$ | 0.61 (0.03) | 0.71 (0.02) | 0.54 (0.01) |
| *folded multi-sample (MD), unfolded multi-sample (MC)* | $-\ln \left\langle \frac{p_\theta(\mathbf{a}'\mid\mathbf{x_a})}{p_\theta(\mathbf{a}\mid\mathbf{x_a})} \right\rangle_F - \ln \left\langle \frac{p_\theta(\mathbf{a}\mid\mathbf{x_a})}{p_\theta(\mathbf{a}'\mid\mathbf{x_a})} \right\rangle_U$ | 0.55 (0.03) | 0.69 (0.02) | 0.53 (0.01) |
| *folded multi-sample, inv-fold single-aa as unfolded* | $-\ln \left\langle \frac{p_\theta(\mathbf{a}'\mid\mathbf{x_a})}{p_\theta(\mathbf{a}\mid\mathbf{x_a})} \right\rangle_F - \ln \frac{p_\theta(\mathbf{a_i}\mid\mathbf{x_{a_{i-1:i+1}}})}{p_\theta(\mathbf{a_i'}\mid\mathbf{x_{a_{i-1:i+1}}})}$ | 0.59 (0.02) | 0.71 (0.02) | 0.53 (0.01) |
| *folded multi-sample (MD), IDP aa-stats as unfolded* | $-\ln \left\langle \frac{p_\theta(\mathbf{a}'\mid\mathbf{x_a})}{p_\theta(\mathbf{a}\mid\mathbf{x_a})} \right\rangle_F - \ln \frac{p_\theta(\mathbf{a}\mid\mathbf{U}_{\mathrm{IDP}})}{p_\theta(\mathbf{a}'\mid\mathbf{U}_{\mathrm{IDP}})}$ | 0.62 (0.03) | 0.72 (0.02) | 0.54 (0.01) |

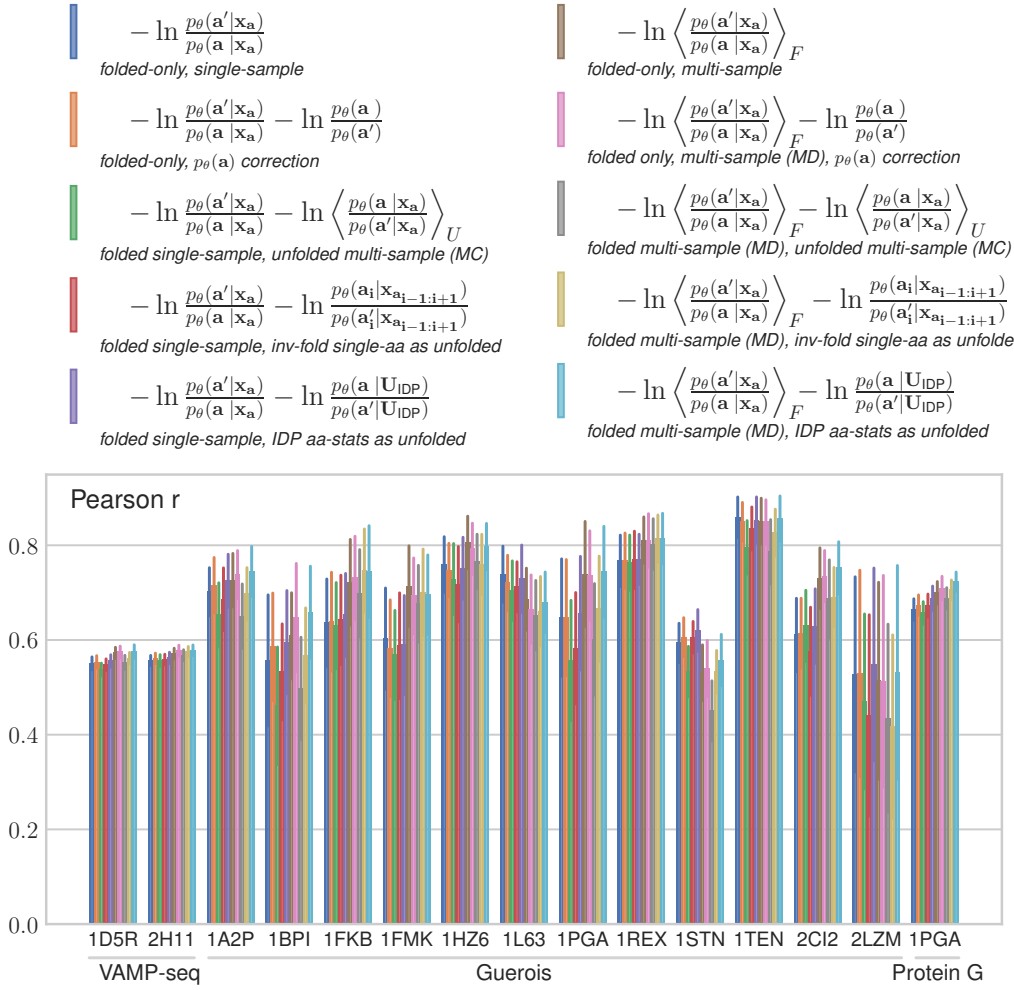

Figure 4: A breakdown of the performance for the individual proteins within the three original datasets. Since correlations are computed, only proteins with at least 20 variant observations are included. The top-left variant is the approach typically employed as zero-shot predictor for protein stability prediction. The left column are methods based that consider only a single folded structure, while the right column considers a structural ensemble from an MD simulation. Note the considerable variation among the proteins in the Guerois set.

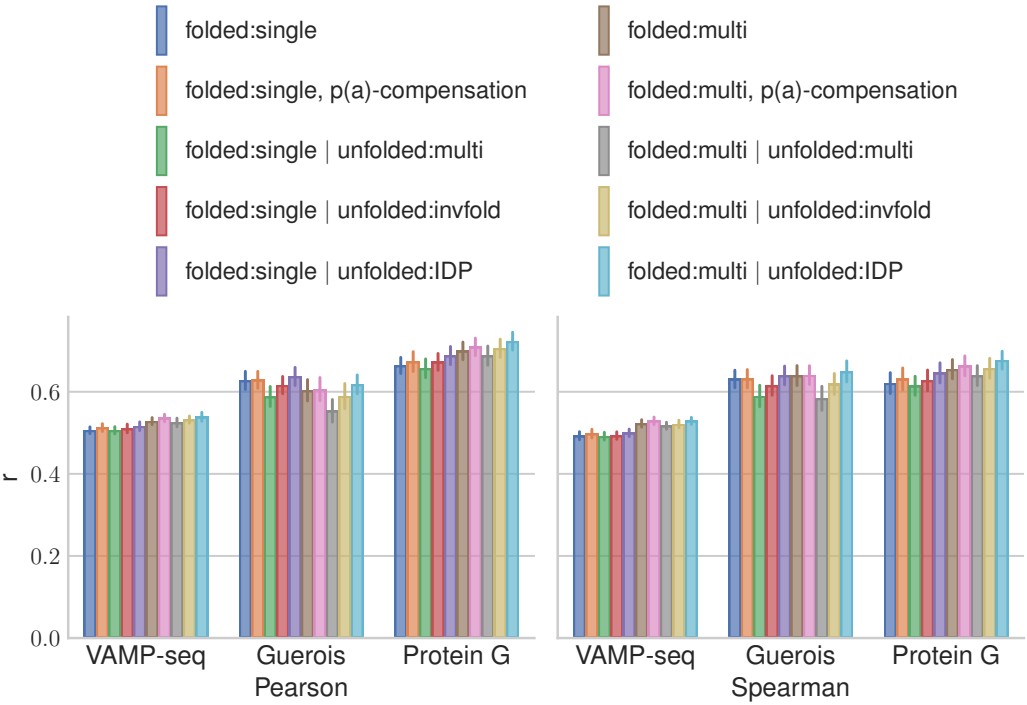

Figure 5: Pearson and Spearman correlations behave similarly. As expected from our derivations, the relationship between zero-shot scores and stability is linear, and employing a rank-based procedure like Spearman rho is therefore not necessary.

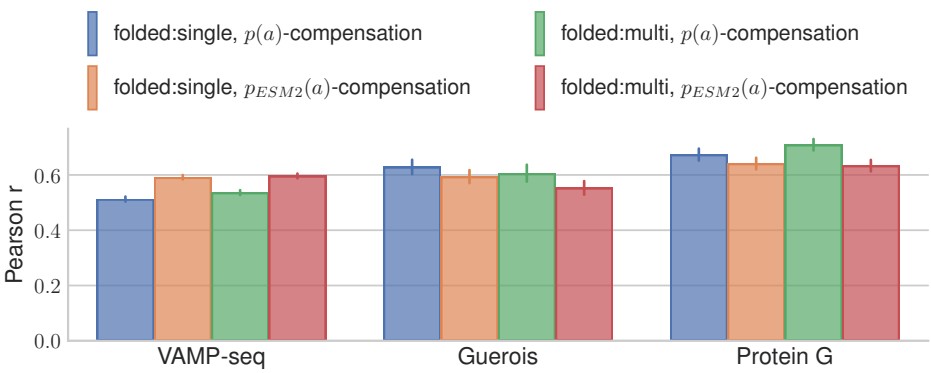

Figure 6: Compensating with a richer model for $p(a)$. A comparison between using the simple site-independent $p(a)$ and a full language model, ESM2 (Lin et al., 2022). The language model does not provide a benefit over the simpler model. One potential explanation could be that the ESM2 model itself captures a considerable structural signal, as illustrated by its use for structure prediction in ESMFold (Lin et al., 2022).

Table 2: Pearson correlation coefficients for different strategies, focusing on the sequence-only strategies (as compensation terms or as a model for the folded state). The first 10 rows in this table are also present in appendix B.1, but are repeated here for ease of comparison. Numbers in parentheses represent standard errors of the mean.

| Strategy | | Guerois | Protein G | VAMP-seq |
|---|---|---|---|---|
| folded:single | | 0.63 (0.02) | 0.66 (0.02) | 0.51 (0.01) |
| | unfolded:multi | 0.59 (0.02) | 0.66 (0.02) | 0.51 (0.01) |
| | unfolded:invfold | 0.62 (0.03) | 0.67 (0.02) | 0.51 (0.01) |
| | unfolded:IDP | 0.64 (0.02) | 0.69 (0.02) | 0.52 (0.01) |
| folded:multi | | 0.60 (0.02) | 0.70 (0.02) | 0.53 (0.01) |
| | unfolded:multi | 0.55 (0.03) | 0.69 (0.02) | 0.53 (0.01) |
| | unfolded:invfold | 0.59 (0.03) | 0.71 (0.02) | 0.53 (0.01) |
| | unfolded:IDP | 0.62 (0.03) | 0.72 (0.02) | 0.54 (0.01) |
| folded:single, $p(a)$-compensation | | 0.63 (0.02) | 0.67 (0.02) | 0.51 (0.01) |
| folded:multi, $p(a)$-compensation | | 0.61 (0.03) | 0.71 (0.02) | 0.54 (0.01) |
| folded:single, $p_{ESM2}(a)$-compensation | | 0.59 (0.03) | 0.64 (0.02) | 0.59 (0.01) |
| folded:multi, $p_{ESM2}(a)$-compensation | | 0.55 (0.03) | 0.63 (0.02) | 0.60 (0.01) |
| folded:$p(a)$ | | 0.05 (0.04) | 0.12 (0.04) | 0.09 (0.01) |
| | unfolded:multi | -0.04 (0.03) | -0.04 (0.04) | 0.05 (0.01) |
| | unfolded:invfold | -0.08 (0.04) | -0.01 (0.04) | 0.01 (0.01) |
| | unfolded:IDP | 0.14 (0.04) | 0.21 (0.03) | 0.12 (0.01) |
| folded:$p_{ESM2}(a)$ | | 0.38 (0.04) | 0.41 (0.03) | 0.49 (0.01) |
| | unfolded:multi | 0.34 (0.03) | 0.37 (0.03) | 0.48 (0.01) |
| | unfolded:invfold | 0.35 (0.03) | 0.39 (0.03) | 0.46 (0.01) |
| | unfolded:IDP | 0.4 (0.03) | 0.45 (0.03) | 0.50 (0.01) |

Table 3: Ablation on the Protein G dataset of 1) the inverse-folding model employed and 2) the procedure for generating structural ensembles. As has been observed previously (Notin et al., 2023), ProteinMPNN generally produces lower correlation scores than ESM-IF, but displays a relatively larger performance boost with the more advanced strategies. Remarkably, the MD and BioEmu results are almost identical on these datasets, suggesting that BioEmu could provide a reasonable approximation to the expensive MD structural ensembles. Values are Pearson correlation scores and standard errors of the mean are provided in parenthesis.

| Strategy | Ensemble type | ESM-IF | MPNN |
|---|---|---|---|
| folded:single | | 0.66 (0.02) | 0.4 (0.03) |
| folded:single, p(a)-compensation | | 0.67 (0.02) | 0.43 (0.03) |
| folded:single \| unfolded:multi | | 0.66 (0.02) | 0.39 (0.03) |
| folded:single \| unfolded:invfold | | 0.67 (0.02) | 0.35 (0.03) |
| folded:single \| unfolded:IDP | | 0.69 (0.02) | 0.48 (0.02) |
| folded:multi | MD | 0.70 (0.02) | 0.50 (0.02) |
| | BioEmu | 0.69 (0.02) | |
| folded:multi, p(a)-compensation | MD | 0.71 (0.02) | 0.53 (0.02) |
| | BioEmu | 0.70 (0.02) | |
| folded:multi \| unfolded:multi | MD | 0.69 (0.02) | 0.49 (0.02) |
| | BioEmu | 0.68 (0.02) | |
| folded:multi \| unfolded:invfold | MD | 0.71 (0.02) | 0.43 (0.03) |
| | BioEmu | 0.70 (0.02) | |
| folded:multi \| unfolded:IDP | MD | 0.72 (0.02) | 0.58 (0.02) |
| | BioEmu | 0.72 (0.02) | |

