# OpenReview forum: "Zero-shot protein stability prediction by inverse folding models: a free energy interpretation"
_NeurIPS.cc/2025/Conference — NeurIPS 2025 poster_

### Official Review · Reviewer_NjCR · 2025-06-28

**Clarity:** 4
**Significance:** 3
**Originality:** 4
**Rating:** 4
**Confidence:** 3

**Summary:**

This paper presents a mathematical framework that ties the ddG of mutation to inverse folding probabilities. More specifically, they show under some assumptions that the ddG can be approximated by the ratio of the inverse folding model's predictions given folded and unfolded structures (Eqn 19).

**Questions:**

Questions
- How does equation 10 connect to the rest of the technical section?
- How does section 3.4 Change in Stability with sequence models tie into estimating protein stability using inverse folding models?
- Why does the inverse folding model compute a meaningful $p_\theta(a|x)$ for unfolded structures, as it was never trained on unfolded structures?
- Could the authors elaborate on the position independent sequence model estimated from $p_D$ described in L248?

**Ethical Concerns:**

["NO or VERY MINOR ethics concerns only"]

**Final Justification:**

my concerns have largely been addressed

**Limitations:**

yes

**Quality:**

3

**Strengths And Weaknesses:**

Strengths
- The technical section is crisp and extremely easy to follow.
- The assumptions made are presented forthright and are largely intuitive and reasonable.

Weaknesses
- Assume the same $\beta$ (and thus temperature) across the data source (PDB).
- The inverse folding model estimates  $p_\theta(a|x)$, however is only trained on folded CATH structures.
- The related work is short and narrowly scoped. It does not discuss other avenues of estimating protein stability beyond inverse folding methods.
	- Tsuboyama, Kotaro, et al. "Mega-scale experimental analysis of protein folding stability in biology and design." Nature 620.7973 (2023).
	- Schymkowitz, Joost, et al. "The FoldX web server: an online force field." Nucleic acids research. (2005).

---

> ### Author Rebuttal · Authors · 2025-07-31
>
> ## Summary of responses to all reviewers
> We thank all the reviewers for their constructive reviews and relevant feedback. The comments regarding the theoretical contribution were generally positive, but several of the reviewers raised concerns regarding the scope of our empirical evaluations. Although we believe that the primary contribution of our paper is the theoretical connection between inverse folding likelihoods and thermodynamic stability, we agree that expanding the empirical evaluations would further strengthen the paper. We have therefore initiated the following experiments:
>
> 1. Repeat experiments with ProteinMPNN, a leading inverse-folding model,
> 2. Replace simplistic site-independent model for $p(\mathbf{a})$ with a pLM,
> 3. Rerun our protocol using ensembles obtained from BioEmu, thereby avoiding the computational bottleneck of molecular dynamics – which would allow us to scale to larger datasets.
>
> We hope to have preliminary results for 1) ready within the rebuttal period, and 2) and 3) will be added before camera-ready. Finally, we will change the paper title to "Zero-shot protein stability prediction by inverse folding models: a free energy interpretation" – to more specifically reflect its contents.
>
> ## Response to Reviewer NjCR
>
>
> We thank Reviewer NjCR for their detailed feedback to our paper. We give a point-by-point response to the identified weaknesses and questions below:
>
> **Assume the same beta (and thus temperature) across the data source (PDB)**
>
> We agree that the assumption about constant $\beta$ (inverse temperature) across all of PDB (eq. 13) is quite strong.  However, we have since realized that this assumption can be substantially relaxed. It is sufficient to assume (cf. eq. 11) that each structure in the database is sampled from its own Boltzmann distribution with a potentially different (and unknown) $\beta$, that is
> $$
> p_{\operatorname{D}}(\boldsymbol{\chi} | \boldsymbol{a}, \beta) \approx p(\boldsymbol{\chi} | \boldsymbol{a}, \beta).
> $$
> Crucially, the downstream analysis does not require access to the individual $\beta$-values. In fact, we can also include the inverse temperature $\beta$ in our model from eq. 12, meaning that we assume a joint model $p_\theta(\mathbf{a}, \mathbf{x}, \beta)$ over sequence, structure and inverse temperature. This weaker assumption above (i.e., allowing for variation in $\beta$ across the database) then implies (cf. eq. 13) that
> $$
>    p_{\theta}(\mathbf{x} | \mathbf{a}, \beta) \approx p_{\operatorname{D}}(\mathbf{x} | \mathbf{a}, \beta) \approx p(\mathbf{x} | \mathbf{a}, \beta) ,
> $$
> Using this assumption, we can now explicitly condition all factors in eq. 17 on the specific $\beta$ beta that the structure is observed and simulated at, and we get
> $$
> \frac{p(\mathbf{x} | \mathbf{a}', \beta)}{p(\mathbf{x} | \mathbf{a}, \beta)} \approx \frac{p_\theta(\mathbf{x} | \mathbf{a}', \beta)}{p_\theta(\mathbf{x} | \mathbf{a}, \beta)}
> = \frac{\frac{p_\theta(\mathbf{a}' | \mathbf{x}, \beta) p_\theta(\mathbf{x}| \beta) } {p_\theta(\mathbf{a}'| \beta)}}{\frac{p_\theta(\mathbf{a} | \mathbf{x}, \beta) p_\theta(\mathbf{x}| \beta) } {p_\theta(\mathbf{a}|\beta)}}
> = \frac{p_\theta(\mathbf{a}' | \mathbf{x}, \beta)}{p_\theta(\mathbf{a} | \mathbf{x}, \beta )} \frac{p_\theta(\mathbf{a}|\beta)}{p_\theta(\mathbf{a}'|\beta)}
> $$
> We now make the same weak assumption that is implicitly made when training an inverse folding model: that, conditioned on a given structure, the distribution over amino acid sequences is independent of the temperature at which the structure was recorded. In other words, we treat the structure as a complete description of the structural state, which means that $p_\theta(\mathbf{a} | \mathbf{x}, \beta) = p_\theta(\mathbf{a} | \mathbf{x})$. We make the similar weak assumption that the sequence distribution is independent of the temperature, i.e., $p_\theta(\mathbf{a} | \beta) = p_\theta(\mathbf{a})$. This gives us that
> $$
> \frac{p(\mathbf{x} | \mathbf{a}', \beta)}{p(\mathbf{x} | \mathbf{a}, \beta)} \approx \frac{p_\theta(\mathbf{a}' | \mathbf{x})}{p_\theta(\mathbf{a} | \mathbf{x})} \frac{p_\theta(\mathbf{a})}{p_\theta(\mathbf{a}')}.
> $$
> This can now be substituted into eq. 16, yielding
> $$
> \beta \Delta \tilde{G}\_{\mathbf{a}' \to \mathbf{a}}^{S} = \ln
> \frac{p(S | \mathbf{a}', \beta)}{p(S | \mathbf{a}, \beta)} \approx \ln \mathbb{E}\_{\mathbf{x} \sim p\_\theta(\mathbf{x} | S, \mathbf{a}, \beta)} \left[ \frac{p\_\theta(\mathbf{a}' | \mathbf{x})}{p\_\theta(\mathbf{a} | \mathbf{x} )} \right] \frac{p\_\theta(\mathbf{a})}{p\_\theta(\mathbf{a}')}  .
> $$
> This provides a temperature-dependent estimate of $\beta \Delta \tilde{G}\_{\mathbf{a}' \to \mathbf{a}}^{S}$, where the temperature is determined by that of the MD simulation, i.e., it is specified in the distribution $p_\theta(\mathbf{x} | S, \mathbf{a}, \beta)$. We have updated the paper to reflect this relaxed assumption.
>
> **The inverse folding model estimates $p(a|x)$, however is only trained on folded CATH structures**
>
> It is indeed not obvious that this is reasonable. Empirically, we found that inverse folding models do seem to provide somewhat reasonable amino acid preferences for unfolded states. A possible explanation is that in the autoregressive factorization of an inverse-folding model, the distribution of the $i$'th amino acid is typically conditioned only on the local structural environment surrounding it. Some of these local structural environments (e.g., coil or surface regions) will be similar to those of unfolded proteins, providing some degree of generalization to the unfolded state. We rely on similar generalizations even within the folded state: if $x$ actually represented the full structural state, CATH would provide us only with a single sequence for $\mathbf{a}$ given $\mathbf{x}$, and our model would predict a delta function for each amino acid, providing no information about alternative amino acids (or $\Delta\Delta G$).
>
> **The related work is short and narrowly scoped. It does not discuss other avenues of estimating protein stability beyond inverse folding methods.**
>
> Due to space constraints, we indeed ended up with a quite narrowly scoped related work section. With the extra page provided for camera-ready, we will broaden it to include a broader overview of the developments over the last decades, including methods such as FoldX and Rosetta.
>
> **How does equation 10 connect to the rest of the technical section?**
>
> Equation 10 is used in 3.3.2. We acknowledge that this was not very clear, since 3.3.2 refers to appendix A.2, which in turn refers to eq 10. We will add a direct reference to eq 10 from 3.3.2 to make the link clearer.
>
> **How does section 3.4 Change in Stability with sequence models tie into estimating protein stability using inverse folding models?**
>
> Section 3.4 was introduced to motivate the way we deal with the unfolded state (using amino acid frequencies from the unfolded state). However, the section indeed suggests a protocol relying entirely on sequence-only models, which we did not find time to test empirically. Since we will be evaluating pLMs for $p(a)$ anyway (see response to reviewer sDYD), we will add the results of this protocol as well. This should integrate section 3.4 more closely with the remaining story.
>
> **Why does the inverse folding model compute a meaningful $p_\theta(a|x)$ for unfolded structures, as it was never trained on unfolded structures?**
>
> See discussion above.
>
> **Could the authors elaborate on the position independent sequence model estimated from  described in L248?**
>
> This is a simple amino acid frequency model estimated from the human proteome (`UP000005640_9606.fasta`). As was also pointed out by reviewer sDYD, it would be meaningful to estimate $p(a)$ using a pLM, and we will include this result in the new version of the manuscript.

---

> > ### Comment · Reviewer_NjCR · 2025-08-06
> >
> > The authors have largely addressed my concerns.
> > Provided that the authors broaden the related works with the extra page provided for camera-ready, I will keep my positive score.

---

### Official Review · Reviewer_r63Z · 2025-06-29

**Clarity:** 3
**Significance:** 2
**Originality:** 2
**Rating:** 4
**Confidence:** 3

**Summary:**

In this paper, the authors lay out a series of assumptions that are needed in order to show that the log-odds based on the inverse-folding model (log p(structure|wild-type seq)/p(structure|mutated seq)) and a small number of (even just one) folded structures is a good approximation to ΔΔG (which is the change in the change of the energy from unfolded to folded states of a sequence after a small mutation.) I enjoyed reading the whole series of approximations, although it raises a question whether those approximations naturally arise from our knowledge of molecular biology and reasonable intuition or were selected to fit this picture that the inverse-folding-based log-odds was found earlier by others as a reasonable approximation to the stability of a protein.

Perhaps, it’s okay as long as we can arrive at a usual practice (the log-odds) from ΔΔG via this chain of assumptions, since this chain of assumptions itself may be what was missing. But, it does raise a question whether we could arrive at this conclusion from a different angle and from a different set of explanations and assumptions. I am curious about such an alternative, since the experimental results show that a supposedly better approximation scheme (under the proposed scheme) to ΔΔG (e.g. by using more folded structures, by drawing unfolded structures, etc.) does not really lead to a significantly better approximation of ΔΔG. the correlation does seem to go up generally, but if you look at the bars carefully in Fig. 1, the improvement is at best modest, and sometimes better approximation schemes result in worse correlations.

So, do I have any potential alternatives in my mind? Unfortunately no.

**Questions:**

See my summary comment above.

**Ethical Concerns:**

["NO or VERY MINOR ethics concerns only"]

**Limitations:**

See my summary comment above.

**Paper Formatting Concerns:**

No concern

**Quality:**

3

**Strengths And Weaknesses:**

See my summary comment above.

---

> ### Author Rebuttal · Authors · 2025-07-31
>
> ## Summary of responses to all reviewers
> We thank all the reviewers for their constructive reviews and relevant feedback. The comments regarding the theoretical contribution were generally positive, but several of the reviewers raised concerns regarding the scope of our empirical evaluations. Although we believe that the primary contribution of our paper is the theoretical connection between inverse folding likelihoods and thermodynamic stability, we agree that expanding the empirical evaluations would further strengthen the paper. We have therefore initiated the following experiments:
>
> 1. Repeat experiments with ProteinMPNN, a leading inverse-folding model,
> 2. Replace simplistic site-independent model for $p(\mathbf{a})$ with a pLM,
> 3. Rerun our protocol using ensembles obtained from BioEmu, thereby avoiding the computational bottleneck of molecular dynamics – which would allow us to scale to larger datasets.
>
> We hope to have preliminary results for 1) ready within the rebuttal period, and 2) and 3) will be added before camera-ready. Finally, we will change the paper title to "Zero-shot protein stability prediction by inverse folding models: a free energy interpretation" – to more specifically reflect its contents.
>
> ## Response to Reviewer r63Z
>
> We thank the Reviewer r63Z for their feedback on our paper. It is a valid point that we cannot rule out that a similar outcome could have been obtained through a different line of reasoning. However, the assumptions we make do not arise out of the blue. They are based on approximations of the underlying biophysical systems. In our view, the value of our work lies in the fact that our derivations make it possible to have very specific discussions about the types of assumptions we make implicitly when using inverse folding models – and potentially develop better models in the future by revisiting these assumptions.
>
> As also discussed in our response to PxEM, we considered our empirical studies primarily as a sanity check on the underlying theory, and were comforted to see that the correction terms generally contribute positively to performance. It is true that many of the correction terms contribute only marginally to the $\Delta\Delta G$ prediction performance. However, apart from the noisy Guerois dataset, we consider the total performance gains as quite substantial (0.66 vs 0.72 for ProteinG and 0.51 vs 0.54 for VAMP-seq), in particular when compared to the small variations typically reported between the top performing methods on ProteinGym.
>
> We acknowledge that stronger conclusions about the relative merits of the individual correction terms would require a more elaborate set of experiments. Scaling up to more systems has been hindered by the fact that we need to conduct full molecular dynamics simulations for each considered system. In an attempt to solve this issue, we are currently investigating the use of BioEmu as a replacement for full molecular dynamics simulations (for details, see the general post above).

---

### Official Review · Reviewer_sDYD · 2025-07-01

**Clarity:** 3
**Significance:** 3
**Originality:** 3
**Rating:** 5
**Confidence:** 4

**Summary:**

The authors investigate the relation between free-energy consideration, from a chemistry perspective, and recent inverse folding models, improving the understanding of good results obtained by these models for stability prediction and testing different formulations (based on different assumptions and data availability) in empirical experiments. The results show that inverse folding models represent a simplistic approximation for the free-energy modeling and different considerations improving the modeling have the potential to improve the performance of free-energy predictors.

**Questions:**

Comments:

1. Equation 22: Should the second term be calculated with the same model as the first term or can different models be used?
2. Can the authors clarify how the first term in equation 25 can be calculated or how this conditioning can be trained?
3. (line 220) The authors mention that they use predicted structures for UR50. Does this mean that some of the structures used for the datasets analyzed are predicted structures?
4. What is the meaning of the sentence in lines 232-235: “While data generated… for protein stability.”
5. (lines 246-249) Why a position-independent sequence model? Why did the authors not test a masked protein LM in this case?
6. The writing and explanation of section 4.3 are confusing. It needs additional explanation, intuition, and, if possible, a visualization. As there is no clear way to do that, readers should be clear about the reasoning of the authors for generating or modeling these unfolded ensembles.
7. Can the authors clarify the differences between this work and the work done by Jiao et al, 2024? (currently being discussed in lines 301-304 and lines 319-321)
8. How can the foundations presented in the work be applied to generate better objectives for self-supervised losses when training models using datasets like the PDB? Do the authors think these can also help model more efficient losses for using datasets with augmented predicted structures?

Minor Comments:

1. Typo: (line 99) “which has by”
2. Typo: (line 120) “change-in-free”
3. Expression: (line 217) “on a representative selection of protein dataset”
4. Typo: Missing a dot in line 271

**Ethical Concerns:**

["NO or VERY MINOR ethics concerns only"]

**Final Justification:**

During the rebuttal phase, the authors clarified my concerns and added more empirical results, which addressed a weakness of the manuscript. My recommended score is based on the theoretical contributions of the proposed manuscript, that is timely in this research field.

**Limitations:**

Yes.

**Paper Formatting Concerns:**

No.

**Quality:**

3

**Strengths And Weaknesses:**

Strengths:

1. The attempt to improve the understanding and the relation between free-energy and inverse folding models is timely, given the results of recent references, and has the potential to improve the pre-training objectives of protein-related foundation models.
2. The authors show the potential of including an unfolded ensemble and approximating the structural ensemble of the folded state with multiple samples.
3. The theoretical derivation is well explained and clear giving meaningful insights for the reader.

Weaknesses:

1. Given the popularity of protein language models, it would improve the strength of the paper using these to model correction terms or terms needing to model p(a).
2. The empirical results hint that a better modeling can improve the performance but a stronger discussion and analysis regarding these empirical results would improve the manuscript. The empirical results are limited in the current version of the manuscript.
3. The clarity of the sections for the folded ensemble and unfolded ensemble could be improved.

---

> ### Author Rebuttal · Authors · 2025-07-31
>
> ## Summary of responses to all reviewers
> We thank all the reviewers for their constructive reviews and relevant feedback. The comments regarding the theoretical contribution were generally positive, but several of the reviewers raised concerns regarding the scope of our empirical evaluations. Although we believe that the primary contribution of our paper is the theoretical connection between inverse folding likelihoods and thermodynamic stability, we agree that expanding the empirical evaluations would further strengthen the paper. We have therefore initiated the following experiments:
>
> 1. Repeat experiments with ProteinMPNN, a leading inverse-folding model,
> 2. Replace simplistic site-independent model for $p(\mathbf{a})$ with a pLM,
> 3. Rerun our protocol using ensembles obtained from BioEmu, thereby avoiding the computational bottleneck of molecular dynamics – which would allow us to scale to larger datasets.
>
> We hope to have preliminary results for 1) ready within the rebuttal period, and 2) and 3) will be added before camera-ready. Finally, we will change the paper title to "Zero-shot protein stability prediction by inverse folding models: a free energy interpretation" – to more specifically reflect its contents.
>
> ## Response to Reviewer sDYD
>
> We thank the Reviewer sDYD for their positive remarks and detailed feedback. We agree that it would be natural to use a language model for calculating $p(\mathbf{a})$, and will add this result to the paper. As stated above in response to reviewers u4zq and PxEM, we also agree that the paper would benefit from a more elaborate empirical evaluation, and we are currently setting up experiments using ProteinMPNN and BioEmu (see details above). Thanks for pointing out the difficulties reading sections 4.2 and 4.3. We will make an effort to rephrase these. And thanks for pointing out the list of typos, which we will incorporate immediately.
>
> Please see the point-by-point answers to the questions below:
>
> 1. *Equation 22: Should the second term be calculated with the same model as the first term or can different models be used?*
> In principle, it should be calculated using the same model. So ideally we would want a joint model that supported marginalizing out the structural context, and we could use the same model for both terms. In practice, inverse-folding models do not support this, and we therefore typically employ different models for these two terms: an inverse-folding model for the first and a sequence-only model for the 2nd.
>
> 2. *Can the authors clarify how the first term in equation 25 can be calculated or how this conditioning can be trained?*
> The first term in equation 25 is the probability of observing an amino acid sequence, conditioned on being in an unfolded state. It is indeed not trivial how to evaluate this in practice – and we therefore dedicate section 4.3 to discussing various alternatives. The most simple solution that we found is to use the amino acid frequencies obtained from intrinsically disordered proteins for this quantity. Given your comment above about section 4.3, we acknowledge that this was not sufficiently clear in the paper, and will elaborate on this explanation in the new version of the paper.
>
> 3. *(line 220) The authors mention that they use predicted structures for UR50. Does this mean that some of the structures used for the datasets analyzed are predicted structures?*
> Apologies for the confusing sentence structure. The intention was to communicate that the original authors of ESM-IF trained their model on both on CATH AND on predicted structures for UR50. When evaluating the models, we only used crystal structures and structures from molecular dynamics simulations. We have rephrased the sentence to avoid the confusion.
>
> 4. *What is the meaning of the sentence in lines 232-235: "While data generated … for protein stability."*
> We apologize for this poorly formulated sentence. We meant to say that the abundance signal measured by VAMP-seq is a convoluted signal of thermodynamic stability, effects on local unfolding stability and interactions with the cell's quality control system, and we therefore do not expect to get as clear a signal on protein stability from this type of experiment. We will rephrase to make this clearer.
>
> 5. *(lines 246-249) Why a position-independent sequence model? Why did the authors not test a masked protein LM in this case?*
> The choice of the position-independent sequence model was made as a simple proxy which had similar sequence statistics as the training data of the ESM-IF model (which was trained on UniRef50 sequences). Protein language models like ESM2 are trained on similar sequence datasets, and should therefore indeed be directly applicable as well. We will do so in the new version of the paper.
>
> 6. *The writing and explanation of section 4.3 are confusing. It needs additional explanation, intuition, and, if possible, a visualization. As there is no clear way to do that, readers should be clear about the reasoning of the authors for generating or modeling these unfolded ensembles.*
> We acknowledge that this was not sufficiently clear. One of the main problems seems to be that we do not explicitly relate the discussion in 4.3 back to the terms we wish to approximate (e.g. the first term in eq 25). We will rework these sections to make this clearer, possibly adding a figure if necessary.
>
> 7. *Can the authors clarify the differences between this work and the work done by Jiao et al, 2024? (currently being discussed in lines 301-304 and lines 319-321)*
> Our work considers the more general case of the full structural ensembles, while Jiao et al. represent structural states by single conformations. In addition, Jiao et al. consider protein binding while we consider protein stability, but as discussed in our response to reviewer u4zq above, these tasks are closely related, and it would be possible to use our approach in the setting introduced by Jiao et al., by considering the full structural ensembles.
>
> 8. *How can the foundations presented in the work be applied to generate better objectives for self-supervised losses when training models using datasets like the PDB? Do the authors think these can also help model more efficient losses for using datasets with augmented predicted structures?*
> Indeed, it would be possible to construct an self-supervised objective, in a very similar way as Jiao et al. (2024), but in our case the loss would be evaluated over a full structural ensemble. It would be interesting to explore whether this leads to improved binding energy prediction. However, due to the considerable experimentation required, we consider it outside the scope of the current manuscript.

---

> > ### Comment · Reviewer_sDYD · 2025-08-01
> > **Response by Reviewer**
> >
> > Thanks for your throughout rebuttal of my comments.
> >
> > The answers by the authors clarified my comments, and I will keep my score.
> >
> > If the results for ProteinMPNN and the modeling of p(a) with a pLM are finished within the Discussion period, I would like to discuss it, given that it would improve the strength of the empirical results presented in the manuscript.

---

### Official Review · Reviewer_PxEM · 2025-07-01

**Clarity:** 3
**Significance:** 1
**Originality:** 2
**Rating:** 4
**Confidence:** 4

**Summary:**

In this paper, the authors investigate the relationship between thermodynamic stability of proteins and log-odds scores from inverse folding models. They derive corrections to the typical simple zero-shot predictor and comprehensively evaluate incorporating additional terms related to the unfolded and folded ensembles. They consider three datasets and conclude that it is possible to improve on the zero-shot predictor by including corrections from the unfolded ensemble and additional samples from the folded ensemble.

**Questions:**

Are there additional datasets or experimental settings, such as subsets of ProteinGym, where the authors think the effects of the corrective terms may be more pronounced?
Can the authors highlight any specific cases where the ranking of different mutants within a dataset via the different methods is substantially different between the different terms considered in the paper?

**Ethical Concerns:**

["NO or VERY MINOR ethics concerns only"]

**Final Justification:**

The authors addressed both my major comments during the rebuttal period, and further clarified the scope and motivation of their work. Hence, I have raised my score.

**Limitations:**

Yes

**Quality:**

2

**Strengths And Weaknesses:**

Strengths
The paper provides a clear and well-reasoned theoretical basis for exploring the connection between thermodynamic stability and likelihoods under an inverse folding model.
The theory naturally suggests two distinct categories of correction terms, and many specific implementations to ablate.
The finding that the unfolded state is best approximated by simple amino acid frequency statistics from disordered regions is interesting and could be of interest for future work on modeling and incorporating information from disordered regions in proteins, which current generation structure-based models largely do not account for.

Weaknesses
The experiments, while thorough and comprehensive across the three datasets considered and the many additional terms, do not suggest any improvements to the folded-only, single-sample standard approach that are robust enough and worth the additional complexity of computing the correction terms.
The paper considers only ESM-IF, whereas most protein design literature relies on ProteinMPNN and related models.
The experiments do not consider more widely used benchmarks like ProteinGym.
Figure 1 is difficult to parse. It may be useful to indicate, via text within the figure and another visual element like hash marks, that the left column considers single-structure approaches and the right column considers ensemble approaches. Highlighting the standard zero-shot predictor would also be helpful.
The abstract and introduction do not mention any specific quantitative results, conclusions, or prescriptions for future work.

---

> ### Author Rebuttal · Authors · 2025-07-31
>
> ## Summary of responses to all reviewers
> We thank all the reviewers for their constructive reviews and relevant feedback. The comments regarding the theoretical contribution were generally positive, but several of the reviewers raised concerns regarding the scope of our empirical evaluations. Although we believe that the primary contribution of our paper is the theoretical connection between inverse folding likelihoods and thermodynamic stability, we agree that expanding the empirical evaluations would further strengthen the paper. We have therefore initiated the following experiments:
>
> 1. Repeat experiments with ProteinMPNN, a leading inverse-folding model,
> 2. Replace simplistic site-independent model for $p(\mathbf{a})$ with a pLM,
> 3. Rerun our protocol using ensembles obtained from BioEmu, thereby avoiding the computational bottleneck of molecular dynamics – which would allow us to scale to larger datasets.
>
> We hope to have preliminary results for 1) ready within the rebuttal period, and 2) and 3) will be added before camera-ready. Finally, we will change the paper title to "Zero-shot protein stability prediction by inverse folding models: a free energy interpretation" – to more specifically reflect its contents.
>
> ## Response to Reviewer PxEM
>
> We thank the Reviewer PxEM for their positive evaluation of our theoretical contribution and their critical assessment of our paper. Our main motivation with this paper was to attempt to understand why log-odds score from inverse-folding models correlate so well with stability changes. From this perspective, we did not consider it a problem that not all correction terms provide substantial improvements, and considered the fact that these correction terms generally had a positive effect mostly as a sanity-check of the underlying derivations.
>
> Of course, we agree with the reviewer that the paper would benefit from an expanded set of experiments – both to get some more statistically significant results and because the assumptions we make might hold to a different degree for different protein systems. As we have discussed above with reviewer u4zq, the main bottleneck was conducting properly converged molecular dynamics simulations. We agree with the reviewer that this computational complexity makes our approach less interesting for practical application. In an effort to overcome this, we are now repeating our experiments using ensembles from the BioEmu model, with the hope that the structural ensembles sampled from this model are of sufficient quality to make it an alternative to molecular dynamics. If this proves successful, it allows us to scale our analysis to a larger stability subset of ProteinGym (e.g., the Tsuboyama subset), and also suggests a path towards making our approach more useful for practical use.
>
> We agree that it makes sense to compare to MPNN as well. We chose ESM-IF because it generally performs slightly better on zero-shot stability prediction than ProteinMPNN (according to ProteinGym), but the reviewer is right that the ubiquitous use of ProteinMPNN makes it a relevant comparison. We are currently running these experiments and hope to have them done before the rebuttal period ends.
>
> We thank the reviewer for their comments regarding Figure 1 and the abstract/introduction, which we will incorporate in the new version of the manuscript.
>
> It is a good question whether subsets of ProteinGym – or differences in experimental settings – would display different tendencies. We could clearly expect the most meaningful results for subsets categorized under the "Stability" label. In particular, the Tsuboyama megascale experiments are direct probes for protein stability and should thus behave well. The VAMP-seq datasets – like the one we use in our paper – measure abundance, but have previously been shown to correlate well with stability, and should thus be another meaningful subset of ProteinGym to consider (although likely displaying somewhat higher levels of noise).
>
> We do see differences in the ranking of mutants between different ablations of terms, as reflected by the Spearman rank correlation scores in Figure 1 and Table 1. It would indeed be interesting to investigate whether there are any patterns to these effects – e.g., whether there is a clear difference between surface and core residues.

---

> > ### Comment · Reviewer_PxEM · 2025-08-05
> >
> > I thank the authors for their response, and acknowledge that the theoretical focus of the paper is well motivated. The updated results with ProteinMPNN and BioEmu address my comments, and I am happy to raise my score.

---

> > > ### Author Response · Authors · 2025-08-08
> > >
> > > Thank you for the feedback – and for increasing your score. We just managed finish the first batch of experiments on the Tsuboyama dataset, using BioEmu ensembles (see global comment), which hopefully provides some confidence that our method could have practical impact as well.

---

### Official Review · Reviewer_u4zq · 2025-07-03

**Clarity:** 3
**Significance:** 3
**Originality:** 3
**Rating:** 5
**Confidence:** 4

**Summary:**

This paper presents a new theoretical perspective on the connection between the likelihood learned by inverse folding models and thermostability. Based on thermodynamic equations, this paper clarifies the free-energy foundations of inverse folding models and explore new ways to estimate relative stability other than the standard log likelihood ratio methods. The experiments on three protein stability datasets show that considerable gains in zero-shot performance can be achieved with fairly simple alternatives.

**Questions:**

1. If you run the same experiments with proteinMPNN, would the performance of different prediction methods change?
2. Can you discuss how to extend your theoretical analysis to binding energy prediction. People typically use the SKEMPI database to evaluate different methods and they found that protein inverse folding models is also predictive. Can we adopt the same extensions you developed here?

**Ethical Concerns:**

["NO or VERY MINOR ethics concerns only"]

**Limitations:**

Yes

**Quality:**

3

**Strengths And Weaknesses:**

Strength:
1. This paper is one of the few papers that presents theoretical perspective on connection between the likelihood learned by inverse folding models and thermostability. It's a very interesting paper. The thermodynamics equations / derivations presented in this paper may inspire other researchers to design better methods for stability prediction
2. Based on the thermodynamic perspective, it proposes new alternatives for stability prediction other than standard log-likelihood ratio. The new alternative metrics actually perform better and more principled.

Weakness
1. The test sets used in this paper is relatively small. Additional datasets can be included, such as Tsuboyama et al. paper listed below.
2. This paper only use ESM-IF. How about proteinMPNN? It's unclear whether the same observation holds for ProteinMPNN.

Tsuboyama K, Dauparas J, Chen J, Laine E, Mohseni Behbahani Y, Weinstein JJ, Mangan NM, Ovchinnikov S, Rocklin GJ. Mega-scale experimental analysis of protein folding stability in biology and design. Nature. 2023
Jul; p. 1–11.

---

> ### Author Rebuttal · Authors · 2025-07-31
>
> ## Summary of responses to all reviewers
> We thank all the reviewers for their constructive reviews and relevant feedback. The comments regarding the theoretical contribution were generally positive, but several of the reviewers raised concerns regarding the scope of our empirical evaluations. Although we believe that the primary contribution of our paper is the theoretical connection between inverse folding likelihoods and thermodynamic stability, we agree that expanding the empirical evaluations would further strengthen the paper. We have therefore initiated the following experiments:
>
> 1. Repeat experiments with ProteinMPNN, a leading inverse-folding model,
> 2. Replace simplistic site-independent model for $p(\mathbf{a})$ with a pLM,
> 3. Rerun our protocol using ensembles obtained from BioEmu, thereby avoiding the computational bottleneck of molecular dynamics – which would allow us to scale to larger datasets.
>
> We hope to have preliminary results for 1) ready within the rebuttal period, and 2) and 3) will be added before camera-ready. Finally, we will change the paper title to "Zero-shot protein stability prediction by inverse folding models: a free energy interpretation" – to more specifically reflect its contents.
>
> ## Response to Reviewer u4zq
>
> We thank Reviewer u4zq for their positive remarks. We agree that the test sets used in our paper were not as extensive as one might have wished for. The primary reason for focusing on a limited set of systems was the molecular dynamics simulations: not only the computational cost, but also making sure that the molecular systems were set up correctly, relaxed properly in the forcefield, etc. We believe the current set of experiments is sufficient as a sanity check of the presented theoretical contribution. However, we do agree that it would be valuable to see whether the trends hold across a larger dataset, and to develop a protocol that would make the approach more robustly applicable in practice. We are therefore now in the process of running our experiments using ensembles sampled from BioEmu, and will attempt to scale this to the Tsuboyama et al dataset. We believe we can have this full evaluation completed in time for camera-ready. If the BioEmu ensemble-approach fails, we would be happy to at least extend our current analysis to several more systems using the full molecular dynamics protocol.
>
> We also agree that it would be interesting to see if MPNN shows similar trends. We are setting up these experiments now. This does not involve new molecular dynamics simulations, we expect that we should have preliminary results ready before the discussion period ends, and hope to post the results when they are ready. A priori, it is unclear to us what to expect from MPNN. The ProteinGym zero-shot benchmark for the stability category reports ESM-IF likelihoods to have somewhat higher correlation to experimental values (0.624) than ProteinMPNN (0.565), suggesting that it might perform slightly worse, but the relative trends of the ablation might show very similar behavior.
>
> It would indeed be interesting to investigate our approach in the context of binding energy. Theoretically, it is a simple extension, since the standard way to calculate mutational effects on binding is to look at the difference in mutational effects on the stability of the complex ($A:B$) and the monomers ($A$ & $B$) is $\Delta\Delta G = \Delta G_{A:B}^{\text{wt} \to \text{mut}} - \Delta G_{A}^{\text{wt} \to \text{mut}} - \Delta G_{B}^{\text{wt} \to \text{mut}}$ . In practice, it would require some work to do the analysis, since structural ensembles would need to be acquired for these different structural states, over a range of systems for which we have experimental results (e.g. from SKEMPI as you suggest). Because of the work involved in these practical matters, we defined this to be out of scope of the current paper, and focused our attention on stability. In the new version of the manuscript, we will, however, add a comment to the discussion to make it clearer how our approach applies to the binding case.

---

### Author Response · Authors · 2025-08-05
**Update on results 1/2**

An update regarding the results: Using **Protein G** as a test case, we have 1) implemented a comparison to **ProteinMPNN**, and 2) conducted experiments replacing our MD ensembles with ensembles from the **BioEmu** generative model.


| Strategy |          | ESM-IF (MD) | MPNN (MD) | ESM-IF (BioEmu) |
|----------|----------|--------|------|--------|
|*folded-only, single-sample* | $-\ln \frac{p\_\theta(\bf{a}' \| \bf{x}\_{\bf{a}})}{p\_\theta(\bf{a}\phantom{'} \| \bf{x}\_{\bf{a}})} \vphantom{\bigg \rangle\_{F}}$ | 0.66 (0.02) | 0.40 (0.03) | 0.66 (0.02) |
*folded-only, $p\_\theta(\bf{a})$ correction* |  $-\ln\frac{p\_\theta(\bf{a}' \| \bf{x}\_{\bf{a}})}{p\_\theta(\bf{a}\phantom{'} \| \bf{x}\_{\bf{a}})} \vphantom{\bigg \rangle_{F}} - \ln \frac{p\_\theta(\bf{a}\phantom{'})}{p\_\theta(\bf{a}')}$  | 0.67 (0.02) | 0.43 (0.02) | 0.67 (0.02)
| *folded single-sample, unfolded multi-sample (MC)* | $-\ln \frac{p\_\theta(\bf{a}' \| \bf{x}\_{\bf{a}})}{p\_\theta(\bf{a}\phantom{'} \| \bf{x}\_{\bf{a}})}  \vphantom{\bigg \rangle\_{F}} - \ln \left \langle \frac{p\_\theta(\bf{a}\phantom{'} \| \bf{x}\_{\bf{a}})}{p\_\theta(\bf{a}' \| \bf{x}\_{\bf{a}})} \right \rangle_{U}$ | 0.66 (0.02) | 0.39 (0.03) | 0.66 (0.02) |
|*folded single-sample, inv-fold single-aa as unfolded* | $-\ln \frac{p\_\theta(\bf{a}' \| \bf{x}\_{\bf{a}})}{p\_\theta(\bf{a}\phantom{'} \| \bf{x}\_{\bf{a}})}  \vphantom{\bigg \rangle\_{F}} - \ln \frac{p\_\theta(\bf{a}\_i \| \bf{x}\_{\bf{a}\_{i-1:i+1}})}{p\_\theta(\bf{a}'\_i \| \bf{x}\_{\bf{a}\_{i-1:i+1}})} \vphantom{\bigg \rangle\_{U}}$ | 0.67 (0.02) | 0.35 (0.03) | 0.67 (0.02) |
|*folded single-sample, IDP aa-stats as unfolded* | $-\ln \frac{p\_\theta(\bf{a}' \| \bf{x}\_{\bf{a}})}{p\_\theta(\bf{a}\phantom{'} \| \bf{x}\_{\bf{a}})}  \vphantom{ \bigg \rangle_{F}} - \ln \frac{p\_\theta(\bf{a}\phantom{'} \| U\_\text{IDP})}{p\_\theta(\bf{a}' \| U\_\text{IDP})}$ | 0.69 (0.02) | 0.48 (0.03) | 0.69 (0.02) |
| *folded-only,* ***multi-sample*** |  $-\ln\left \langle\frac{p\_\theta(\bf{a}' \| \bf{x}\_{\bf{a}})}{p\_\theta(\bf{a}\phantom{'} \| \bf{x}\_{\bf{a}})}\right \rangle\_{F}$  | 0.70 (0.02) | 0.50 (0.02) | 0.69 (0.02) |
|*folded only*, ***multi-sample***, *$p\_\theta(\bf{a})$ correction* |   $-\ln \left \langle \frac{p\_\theta(\bf{a}' \| \bf{x}\_{\bf{a}})}{p\_\theta(\bf{a}\phantom{'} \| \bf{x}\_{\bf{a}})} \right \rangle\_{F} - \ln \frac{p\_\theta(\bf{a}\phantom{'})}{p\_\theta(\bf{a}')}$ | 0.71 (0.02) | 0.53 (0.03) | 0.70 (0.02) |
| *folded* ***multi-sample***, *unfolded multi-sample (MC)* | $-\ln \left \langle \frac{p\_\theta(\bf{a}' \| \bf{x}\_{\bf{a}})}{p\_\theta(\bf{a}\phantom{'} \| \bf{x}\_{\bf{a}})} \right \rangle\_{F} - \ln \left \langle \frac{p\_\theta(\bf{a}\phantom{'} \| \bf{x}\_{\bf{a}})}{p\_\theta(\bf{a}' \| \bf{x}\_{\bf{a}})} \right \rangle\_{U}$ | 0.69 (0.02) | 0.49 (0.03) | 0.68 (0.02) |
| *folded* ***multi-sample***, *inv-fold single-aa as unfolded* | $-\ln \left \langle \frac{p\_\theta(\bf{a}' \| \bf{x}\_{\bf{a}})}{p\_\theta(\bf{a}\phantom{'} \| \bf{x}\_{\bf{a}})} \right \rangle\_{F} - \ln \frac{p\_\theta(\bf{a}\_i \| \bf{x}\_{\bf{a}\_{i-1:i+1}})}{p\_\theta(\bf{a}'\_i \| \bf{x}\_{\bf{a}\_{i-1:i+1}})} \vphantom{\bigg \rangle\_{U}}$ | 0.71 (0.03) | 0.43 (0.03) | 0.70 (0.02) |
| *folded* ***multi-sample***, *IDP aa-stats as unfolded* | $-\ln \left \langle \frac{p\_\theta(\bf{a}' \| \bf{x}\_{\bf{a}})}{p\_\theta(\bf{a}\phantom{'} \| \bf{x}\_{\bf{a}})} \right \rangle\_{F}  - \ln \frac{p\_\theta(\bf{a}\phantom{'} \| U\_\text{IDP})}{p\_\theta(\bf{a}' \| U\_\text{IDP})}$ | 0.72 (0.02) | 0.58 (0.02) | 0.72 (0.02) |

The ProteinMPNN results show what we had anticipated: that the overall correlation coefficients are lower (as reported earlier in ProteinGym), but that we see similar increasing trends in the ablations. **The improvements across ablations for ProteinMPNN are actually more striking than they were for ESM-IF, increasing from 0.40 to 0.58 in Pearson r**.

The BioEmu results are almost identical to the original MD results. This is interesting, as it suggests that for this setting, molecular simulations can potentially be meaningfully replaced by faster predicted ensembles. We used the ColabFold implementation of BioEmu with default settings, which completed in a few minutes. This means that our best-performing protocol - combining a folded ensemble with unfolded IDP statistics - could be a practically relevant algorithm (addressing the comment raised by reviewer PxEM).

---

> ### Author Response · Authors · 2025-08-05
> **Update on results 2/2**
>
> The encouraging results from the BioEmu experiments make it possible to scale our experiments across a larger range of proteins. We are now setting up experiments on a 95-protein subset of the Tsuboyama dataset, which was withheld as a validation set by the authors of the BioEmu paper [1], and should thus be a fair way to test the effectiveness of predicted ensembles in our $\Delta\Delta G$-protocol. It is unlikely that all these runs will be finished before the discussion period ends, but we'll post any updates we have prior to the discussion closing.
>
> While still preliminary, we hope the results above provide the reviewers with additional evidence of the practical utility of our contribution and the robustness of our approach to 1) using alternative inverse-folding methods and 2) employing different procedures for generating structural ensembles.
>
> Ultimately, the main contribution of our paper is theoretical, establishing a formal connection between likelihood ratios of inverse folding models and protein relative stabilities. The theoretical part seems to have been well received by all reviewers, but if anyone has any remaining concerns they would like to discuss, please let us know.
>
> [1] Lewis, S., Hempel, T., Jiménez-Luna, J., Gastegger, M., Xie, Y., Foong, A.Y., Satorras, V.G., Abdin, O., Veeling, B.S., Zaporozhets, I. and Chen, Y., 2025. Scalable emulation of protein equilibrium ensembles with generative deep learning. Science, p.eadv9817.

---

### Author Response · Authors · 2025-08-08
**A final status on the requested additional experiments 1/2**

# Protein language model for $p\_\theta(\mathbf{a})$

In response to Reviewers sDYD and NjCR, we have conducted an experiment of replacing the site-independent $p_\theta(\mathbf{a})$ model with **the ESM2 language model on Protein G**. The experiments were conducted using both ESM-IF (MD and BioEmu) and ProteinMPNN (using MD). For reference, we included the previous results on Protein G.

| Strategy | | ESM-IF (MD) | MPNN (MD) | ESM-IF (BioEmu) |
|----------|-|-------------|-----------|-----------------|
|*folded-only, single-sample*|  $-\ln \frac{p\_\theta(\bf{a}' \| \bf{x}\_{\bf{a}})}{p\_\theta(\bf{a} \| \bf{x}\_{\bf{a}})}$ | 0.66 (0.02) | 0.40 (0.03) | 0.66 (0.02) |
|*folded-only, $p\_\theta(\bf{a})$ correction*|  $-\ln\frac{p\_\theta(\bf{a}' \| \bf{x}\_{\bf{a}})}{p\_\theta(\bf{a} \| \bf{x}\_{\bf{a}})} - \ln \frac{p\_\theta(\bf{a})}{p\_\theta(\bf{a}')}$  | 0.67 (0.02) | 0.43 (0.02) | 0.67 (0.02) |
|*folded single-sample, unfolded multi-sample (MC)*| $-\ln \frac{p\_\theta(\bf{a}' \| \bf{x}\_{\bf{a}})}{p\_\theta(\bf{a} \| \bf{x}\_{\bf{a}})}  - \ln \left \langle \frac{p\_\theta(\bf{a} \| \bf{x}\_{\bf{a}})}{p\_\theta(\bf{a}' \| \bf{x}\_{\bf{a}})} \right \rangle\_{U}$ | 0.66 (0.02) | 0.39 (0.03) | 0.66 (0.02) |
|*folded single-sample, inv-fold single-aa as unfolded*| $-\ln \frac{p\_\theta(\bf{a}' \| \bf{x}\_{\bf{a}})}{p\_\theta(\bf{a} \| \bf{x}\_{\bf{a}})}  - \ln \frac{p\_\theta(\bf{a}\_i \| \bf{x}\_{\bf{a}\_{i-1:i+1}})}{p\_\theta(\bf{a}'\_i \| \bf{x}\_{\bf{a}\_{i-1:i+1}})} $ | 0.67 (0.02) | 0.35 (0.03) | 0.67 (0.02) |
 |*folded single-sample, IDP aa-stats as unfolded*|   $-\ln \frac{p\_\theta(\bf{a}' \| \bf{x}\_{\bf{a}})}{p\_\theta(\bf{a} \| \bf{x}\_{\bf{a}})} - \ln \frac{p\_\theta(\bf{a} \| U\_\text{IDP})}{p\_\theta(\bf{a}' \| U\_\text{IDP})} $ | 0.69 (0.02) | 0.48 (0.03) | 0.69 (0.02) |
 | | | | | |
|*folded-only,* ***multi-sample***|  $-\ln\left \langle\frac{p\_\theta(\bf{a}' \| \bf{x}\_{\bf{a}})}{p\_\theta(\bf{a} \| \bf{x}\_{\bf{a}})}\right \rangle\_{F}$  | 0.70 (0.02) | 0.50 (0.02) | 0.69 (0.02) |
 |*folded only,* ***multi-sample***, *$p\_\theta(\bf{a})$ correction*|   $-\ln \left \langle \frac{p\_\theta(\bf{a}' \| \bf{x}\_{\bf{a}})}{p\_\theta(\bf{a} \| \bf{x}\_{\bf{a}})} \right \rangle\_{F} - \ln \frac{p\_\theta(\bf{a})}{p\_\theta(\bf{a}')}$ | 0.71 (0.02) | 0.53 (0.03) | 0.70 (0.02) |
|*folded* ***multi-sample***, *unfolded multi-sample (MC)*| $-\ln \left \langle \frac{p\_\theta(\bf{a}' \| \bf{x}\_{\bf{a}})}{p\_\theta(\bf{a} \| \bf{x}\_{\bf{a}})} \right \rangle\_{F} - \ln \left \langle \frac{p\_\theta(\bf{a} \| \bf{x}\_{\bf{a}})}{p\_\theta(\bf{a}' \| \bf{x}\_{\bf{a}})} \right \rangle\_{U}$ | 0.69 (0.02) | 0.49 (0.03) | 0.68 (0.02) |
|*folded* ***multi-sample***, *inv-fold single-aa as unfolded*| $-\ln \left \langle \frac{p\_\theta(\bf{a}' \| \bf{x}\_{\bf{a}})}{p\_\theta(\bf{a} \| \bf{x}\_{\bf{a}})} \right \rangle\_{F} - \ln \frac{p\_\theta(\bf{a}\_i \| \bf{x}\_{\bf{a}\_{i-1:i+1}})}{p\_\theta(\bf{a}'\_i \| \bf{x}\_{\bf{a}\_{i-1:i+1}})} $  | 0.71 (0.03) | 0.43 (0.03) | 0.70 (0.02) |
 |*folded* ***multi-sample***, *IDP aa-stats as unfolded*| $-\ln \left \langle \frac{p\_\theta(\bf{a}' \| \bf{x}\_{\bf{a}})}{p\_\theta(\bf{a} \| \bf{x}\_{\bf{a}})} \right \rangle\_{F} - \ln \frac{p\_\theta(\bf{a} \| U\_\text{IDP})}{p\_\theta(\bf{a}' \| U\_\text{IDP})}$ | 0.72 (0.02) | 0.58 (0.02) | 0.72 (0.02) |
| | | | | |
|*LLM $p\_\text{ESM2}(\mathbf{a})$ ratio only*| $\ln \frac{p\_\text{ESM2}(\bf{a})}{p\_\text{ESM2}(\bf{a}')}$ | 0.41 (0.03) | 0.41 (0.03) | 0.41 (0.03) |
|*folded-only, $p\_\text{ESM2}(\bf{a})$ correction*| $-\ln\frac{p\_\theta(\bf{a}' \| \bf{x}\_{\bf{a}})}{p\_\theta(\bf{a} \| \bf{x}\_{\bf{a}})} - \ln \frac{p\_\text{ESM2}(\bf{a})}{p\_\text{ESM2}(\bf{a}')}$  | 0.64 (0.02) | 0.48 (0.03) | 0.64 (0.02) |
|*folded only, multi-sample, $p\_\text{ESM2}(\bf{a})$ correction*|  $-\ln \left \langle \frac{p\_\theta(\bf{a}' \| \bf{x}\_{\bf{a}})}{p\_\theta(\bf{a} \| \bf{x}\_{\bf{a}})} \right \rangle\_{F} - \ln \frac{p\_\text{ESM2}(\bf{a})}{p\_\text{ESM2}(\bf{a}')}$ | 0.63 (0.02) | 0.5 (0.02) | 0.62 (0.02) |
|*LLM $p\_{\text{ESM2}}(a)$ ratio, IDP aa-stats as unfolded*|  $\ln \frac{p\_{\text{ESM2}}(\bf{a})}{p\_{\text{ESM2}}(\bf{a}')} - \ln \frac{p\_\theta(\bf{a} \| U\_\text{IDP})}{p\_\theta(\bf{a}' \| U\_\text{IDP})}$  | 0.45 (0.03) | 0.45 (0.03) | 0.45 (0.03) |

The more powerful sequence model does not seem to translate into a significant improvement in $\Delta\Delta G$ correlation (slightly better for MPNN, slightly worse for ESM-IF). However, the LLM results also allow us to make a stronger connection in our paper to section 3.4 (addressing a point made by NjCR) – by considering the setting where a pure sequence model is corrected with a model for the unfolded state (last row in the table above). Interestingly, this indeed seems to be an improvement over using raw ESM2 likelihood ratios. We will expand these results to the rest of the experimental systems in the final version of the paper.

---

> ### Author Response · Authors · 2025-08-08
> **A final status on the requested additional experiments 2/2**
>
> # Tsuboyama dataset
>
> Regarding the scaling of our analysis to more systems, we also managed to complete ten out of the ~40 runs **on the Tsuboyama dataset (from ProteinGym), using BioEmu to generate the structural ensembles**.
>
> | Strategy | | Pearson r (ESM-IF & BioEmu)|
> |----------|-|----------|
> |*folded-only, single-sample*| $-\ln \frac{p\_\theta(\bf{a}' \| \bf{x}\_{\bf{a}})}{p\_\theta(\bf{a} \| \bf{x}\_{\bf{a}})}$ | 0.66 (0.01) |
> |*folded-only, $p\_\theta(\bf{a})$ correction*| $-\ln\frac{p\_\theta(\bf{a}' \| \bf{x}\_{\bf{a}})}{p\_\theta(\bf{a} \| \bf{x}\_{\bf{a}})} - \ln \frac{p\_\theta(\bf{a})}{p\_\theta(\bf{a}')}$ | 0.68 (0.00) |
> |*folded single-sample, IDP aa-stats as unfolded*| $-\ln \frac{p\_\theta(\bf{a}' \| \bf{x}\_{\bf{a}})}{p\_\theta(\bf{a} \| \bf{x}\_{\bf{a}})} - \ln \frac{p\_\theta(\bf{a} \| U\_\text{IDP})}{p\_\theta(\bf{a}' \| U\_\text{IDP})} $ | 0.70 (0.01) |
> |*folded-only, multi-sample*| $-\ln\left \langle\frac{p\_\theta(\bf{a}' \| \bf{x}\_{\bf{a}})}{p\_\theta(\bf{a} \| \bf{x}\_{\bf{a}})}\right \rangle\_{F}$ | 0.68 (0.01) |
> |*folded only, multi-sample (MD), $p\_\theta(\bf{a})$ correction*| $-\ln \left \langle \frac{p\_\theta(\bf{a}' \| \bf{x}\_{\bf{a}})}{p\_\theta(\bf{a} \| \bf{x}\_{\bf{a}})} \right \rangle\_{F}  - \ln \frac{p\_\theta(\bf{a})}{p\_\theta(\bf{a}')}$ | 0.72 (0.00) |
> |*folded multi-sample (MD), IDP aa-stats as unfolded*| $-\ln \left \langle \frac{p\_\theta(\bf{a}' \| \bf{x}\_{\bf{a}})}{p\_\theta(\bf{a} \| \bf{x}\_{\bf{a}})} \right \rangle\_{F}  - \ln \frac{p\_\theta(\bf{a} \| U\_\text{IDP})}{p\_\theta(\bf{a}' \| U\_\text{IDP})}$ | 0.73 (0.00) |
>
> The results look similar to the performances we reported for Protein G in the submitted version of the paper, with a consistent and significant increase in performance both when taking into account the structural ensemble (now through BioEmu) and when modelling the unfolded state (using IDP statistics). We will add a comprehensive overview and discussion of these new results in the paper (once the remaining experiments have been completed).
>
> **Thanks to the reviewers for their constructive feedback – we believe the additional experiments have significantly improved the empirical part of the paper, clearly showing the theoretical considerations to have significant practical effects as well.**

---

### Note · Authors · 2025-08-12

We thank the reviewers for their positive evaluation and constructive feedback. The main points raised by the reviewers concerned:

1. The assumption of constant temperature across PDB entries.
2. Extending the empirical evaluation to ProteinMPNN.
3. The practical applicability of our approach, given the requirement of conducting MD simulations.
4. Evaluating on additional datasets (e.g., Tsuboyama et al.).
5. Replacing the site-independent $p(\textbf{a})$ model with a protein language model.

We **addressed all these concerns** during the rebuttal period:

1. **Theory**: We extended the theory and **relaxed the constant-temperature assumption**, introducing a formulation that conditions on the specific temperature for each structure.
2. **ProteinMPNN**: Experiments showed similar trends to ESM-IF, with **even larger improvements** across ablations (Pearson r rising from 0.40 to 0.58).
3. **Practical applicability:** Replacing MD ensembles with **BioEmu** ensembles produced **near-identical results**, suggesting that predicted ensembles can serve as an efficient substitute for molecular dynamics in our setting.
4. **Additional dataset (Tsuboyama)**: On a subset of the **Tsuboyama dataset** (from ProteinGym) using BioEmu ensembles, we observed performance patterns similar to those for Protein G, with consistent and significant gains when incorporating structural ensembles and unfolded-state modelling.
5. **Protein language model for $p(\textbf{a})$**: Substituting **ESM2** for the site-independent model gives only minor changes in correlation, but provides a stronger conceptual link to Section 3.4.

These extensions strengthen both the theoretical and empirical parts of the paper, and show that the theoretical considerations have clear practical implications. We thank the reviewers again for their suggestions, which have led to a substantially improved final version.

---

### Decision · Program_Chairs · 2025-09-17

**Decision:**

Accept (poster)

**Comment:**

The reviewers and I are in unanimous agreement on accepting the paper: it makes an insightful connection between thermodynamic stability and inverse folding probabilities and experimentally explores how this results in corrections to the zero-shot predictor using unfolded and folded ensembles. Initially multiple reviewers found only using ESM-IF insufficient, but the rebuttal adding ProteinMPNN convincingly addresses this.